# Cross-Domain Off-Policy Evaluation and Learning for Contextual Bandits

**Yuta Natsubori**
Hakuhodo DY Holdings, Inc.
`yuta.natsubori@hakuhodo.co.jp`

**Masataka Ushiku**
Hakuhodo DY Holdings, Inc.
`masataka.ushiku@hakuhodo.co.jp`

**Yuta Saito**
Cornell University
`ys522@cornell.edu`

## Abstract

Off-Policy Evaluation and Learning (OPE/L) in contextual bandits is rapidly gaining popularity in real systems because new policies can be evaluated and learned securely using only historical logged data. However, existing methods in OPE/L cannot handle many challenging but prevalent scenarios such as few-shot data, deterministic logging policies, and new actions. In many applications, such as personalized medicine, content recommendations, education, and advertising, we need to evaluate and learn new policies in the presence of these challenges. Existing methods cannot evaluate and optimize effectively in these situations due to the notorious variance issue or limited/no exploration in the logged data. To enable OPE/L even under these unsolved challenges, we propose a new problem setup of *Cross-Domain OPE/L*, where we have access not only to the logged data from the *target domain* in which the new policy will be implemented but also to logged datasets collected from other domains. This novel formulation is widely applicable because we can often use historical data not only from the target hospital, country, device, or user segment but also from other hospitals, countries, devices, or segments. We develop a new estimator and policy gradient method to solve OPE/L by leveraging both target and source datasets, resulting in substantially enhanced OPE/L in the previously unsolved situations in our empirical evaluations.

## 1 Introduction

Many decision-making systems (e.g., recommendation, medication, budget allocation) interact with the environment through a contextual bandit process in which a policy observes the context, takes action, and obtains rewards. Off-Policy Evaluation and Learning (OPE/L) has gained attention as a technique for estimating and learning new policies without deploying them, using only historical logged data. We can find many real applications of these techniques since they do not require costly and risky online A/B testing and exploration to enable the data-driven policy evaluation and learning lifecycle (Mehrotra et al., 2018; Gilotte et al., 2018; Saito et al., 2021; Kiyohara et al., 2024a).

While recent advances have led to the development of a number of estimators and policy gradient methods (Saito & Joachims, 2021; Uehara et al., 2022), most of these are based on inverse propensity scoring (IPS), reward regression, or their mixture (Dudík et al., 2014; Wang et al., 2017; Su et al., 2020a). These estimators heavily rely on a theoretical assumption called *common support* to provide a low-bias estimate. Due to the common support assumption, we can only evaluate and learn new policies regarding actions that have already been sufficiently explored by the logging policy (Sachdeva et al., 2020; Felicioni et al., 2022). Therefore, in challenging but realistic cases where the logging policy is completely deterministic or there are new actions, existing methods simply cannot evaluate and choose under-explored and new actions at all due to the lack of their reward information in the historical data (Sachdeva et al., 2020). In addition, the use of importance weighting often causes severe variance issues, particularly when the sample size is small, such as *few-shot* data and the action space is large (Saito & Joachims, 2022; Cief et al., 2024a; Sachdeva et al., 2024).

Figure 1: Comparison of **Conventional OPE/L** and **Cross-Domain OPE/L (Ours)**.

There have been several previous efforts to enable OPE (Felicioni et al., 2022) and OPL (Sachdeva et al., 2020) under the violation of common support (also known as support deficiency), but they cannot handle deterministic logging policies and completely new actions. It might also be useful to use some structure in the action or reward space to relax the requirement of common support, as studied by Saito & Joachims (2022); Saito et al. (2023); Cief et al. (2024a); Taufiq et al. (2023); Sachdeva et al. (2024); Kiyohara et al. (2024b), but this useful structure is not always learnable.

To enable effective OPE/L even with deterministic logging policies and new actions, we propose a new problem formulation called ***Cross-Domain OPE/L*** where we aim to evaluate and optimize the value of new policies under the target domain, but have access to historical logged data collected previously in source domains (as depicted in Figure 1). There are many cases where we have access to such multiple-logged datasets. In the medical field, an example scenario is when the impact of a treatment on a patient's prognosis is recorded in several hospitals of different sizes and with different patient demographics. Another example is in recommender systems where new content or function may become available in certain countries or for a subset of users (e.g., active members), and these can be source domains when we aim to perform OPE/L regarding new countries and other users, such as relatively new members. If we have access to these logged datasets collected in source domains with more data and exploration in the action space, we can leverage these data to perform OPE/L more effectively towards the target domain even when it has less logged data, a deterministic logging policy, and new actions that existing estimators cannot address.

After formally presenting the new formulation, we propose a new estimator, called ***Cross-domain Off-Policy Evaluation (COPE)*** and respective policy gradient method to solve OPE/L by effectively leveraging useful information from source domains to estimate and optimize the value of new policies in the target domain. To achieve effective transfer from source domains, our methods are based on a decomposition of the expected reward function into the domain-cluster effect and domain-specific effect. The domain-cluster effect is a component in the reward function that domains in the same cluster have in common, while the domain-specific effect represents the causal effect not modeled merely by the domain-cluster effect. The COPE estimator unbiasedly estimates the domain-cluster effect by applying *multiple importance weighting* (Owen, 2013; Agarwal et al., 2017) using data not only from the target domain but also leveraging data from the source domains that are in the same cluster as the target domain. It also deals with the domain-specific effect via reward regression using the logged data from the target domain, which reduces the bias of the estimator depending on the regression accuracy. We also extend the COPE estimator as a policy gradient estimator to perform OPL using both target and source data, enabling us to perform policy learning even under completely deterministic logging and new actions involved. Theoretical analysis shows that COPE has analyzable bias and, in particular, that COPE can be unbiased even when the target domain has a deterministic logging policy and new actions. Empirical evaluation using a real-world recommendation dataset demonstrates that COPE outperforms existing estimators and policy learning methods by properly leveraging data from both the target and source domains, particularly when there are few target data and many actions that are not previously explored in the target domain.

**Key Related Work.** This section differentiates our work from two closely related studies. A more comprehensive overview of related work is available in Appendix A.

First, we discuss the work by Uehara et al. (2020), which studied OPE/L under covariate shift, where the context distributions differ between the logged data ($p^{\text{hist}}(x)$) and the evaluation environment ($p^{\text{eval}}(x)$), while the reward distribution $p(r|x, a)$ remains unchanged. Under this setup, Uehara et al. (2020) proposes methods to estimate the value of new policies when deployed in the evalua-

tion environment, using only the logged data collected under the historical distribution $p^{\text{hist}}(x)$. In contrast, we aim to estimate the same estimand, but leveraging data from both target and source domains (where there could be multiple source domains), with each domain having its own unique context and reward distributions. By leveraging this new setup, our primary goal is to address challenging scenarios in the target domain, such as new actions, deterministic logging, and extremely small logged data, which clearly differ from the motivations of Uehara et al. (2020).

We also distinguish our contributions from those of Saito et al. (2023), which developed the OffCEM estimator to handle OPE in large action spaces within a single domain setup. Although our main idea of reward function decomposition is inspired by OffCEM, its application to solve the problem of Cross-Domain OPE/L would not be possible without our unique formulation. Moreover, we propose an extension of our estimator to an OPL method, whereas Saito et al. (2023) focused solely on the OPE problem. Therefore, our work is the first to formulate the problem of cross-domain OPE/L and leverage the respective version of reward function decomposition to solve non-trivial challenges, such as new actions and deterministic logging, and thus offers several unique contributions from both methodological and empirical angles.

Finally, we discuss an important distinction between our work and those that address limited overlap or deficient support issues (Hansen, 2008; Sachdeva et al., 2020; Wu & Fukumizu, 2021; Felicioni et al., 2022). Limited overlap refers to a situation where certain actions that can be taken under a new policy have zero probability of being observed under the logging policy. Limited overlap is problematic because, without any data about actions in the logged data, importance weighting techniques become biased. It is important to note that our work aims to address even more challenging scenarios compared to the general limited overlap problem, namely, completely deterministic logging and new actions. Deterministic logging refers to a situation where the logging policy selects a specific action with probability one, i.e., there is no stochasticity. This is more difficult because the typical limited overlap issue still allows the logging policy to be stochastic. Completely new actions present an even greater challenge, as they refer to actions $a$ that have zero probability of being observed in the logged data for any context. To the best of our knowledge, no previous work specifically addresses the issues of completely deterministic logging and new actions. We tackle these extremely challenging scenarios by newly formulating the problem of cross-domain OPE/L.

## 2 CONVENTIONAL FORMULATION

We begin with describing the conventional formulation of OPE in the contextual bandit setting. Here, a decision maker repeatedly observes a context $x \in \mathcal{X}$ drawn from an unknown distribution $p(x)$. Given a context $x$, a fixed and potentially stochastic policy $\pi(a|x)$ selects an action $a$ from a finite action space denoted by $\mathcal{A}$. Reward $r$ is then observed following an unknown distribution $p(r|x,a)$ and we use $q(x,a) := \mathbb{E}[r|x,a]$ to denote the expected reward given context and action. We define the performance measure of policy $\pi$ by the expected reward under its deployment as

$$V(\pi) := \mathbb{E}_{p(x)\pi(a|x)p(r|x,a)}[r] = \mathbb{E}_{p(x)\pi(a|x)}[q(x,a)], \tag{1}$$

which is often called the *policy value*. The logged bandit data we can use to perform OPE/L can be denoted by $\mathcal{D} := \{(x_i, a_i, r_i)\}_{i=1}^n$, which contains $n$ independent observations drawn from the data distribution induced by the logging policy $\pi_0$, i.e., $p(\mathcal{D}) = \prod_{i=1}^n p(x_i)\pi_0(a_i|x_i)p(r_i|x_i, a_i)$.

The aim of OPE is to develop an estimator $\hat{V}$ that can accurately estimate the policy value of a new policy $\pi$ using only $\mathcal{D}$. We measure the accuracy of $\hat{V}$ by the mean squared error (MSE) defined as

$$\text{MSE}(\hat{V}(\pi; \mathcal{D})) := \mathbb{E}_{p(\mathcal{D})}[(V(\pi) - \hat{V}(\pi; \mathcal{D}))^2] = \text{Bias}(\hat{V}(\pi; \mathcal{D}))^2 + \mathbb{V}_{\mathcal{D}}[(\hat{V}(\pi; \mathcal{D}))],$$

where $\mathbb{E}_{p(\mathcal{D})}[\cdot]$ takes the expectation over the distribution of $\mathcal{D}$.

**Limitations of Existing Estimators.** As an existing method for OPE, we first describe IPS (Horvitz & Thompson, 1952), which estimates the policy value by re-weighting the rewards as

$$\hat{V}_{\text{IPS}}(\pi; \mathcal{D}) := \frac{1}{n} \sum_{i=1}^n \frac{\pi(a_i|x_i)}{\pi_0(a_i|x_i)} r_i = \frac{1}{n} \sum_{i=1}^n w(x_i, a_i)r_i, \tag{2}$$

where $w(x,a) := \pi(a|x)/\pi_0(a|x)$ is called the *importance weight*. It is widely known that IPS is unbiased, i.e., $\mathbb{E}_{p(\mathcal{D})}[\hat{V}_{\text{IPS}}(\pi; \mathcal{D})] = V(\pi)$, under the common support condition.

**Condition 2.1** (Common Support). The logging policy $\pi_0$ is said to have common support for policy $\pi$ if $\pi(a|x) > 0 \implies \pi_0(a|x) > 0$ for all $a \in \mathcal{A}$ and $x \in \mathcal{X}$.

This assumption for unbiasedness of IPS is never satisfied in situations when the logging policy $\pi_0$ is deterministic or there are new actions that were not available when the logged data was collected. Under the violation of Condition 2.1, IPS produces the following bias (Sachdeva et al., 2020)

$$|\text{Bias}(\hat{V}_{\text{IPS}}(\pi; \mathcal{D}))| = \mathbb{E}_{p(x)}\left[\sum_{a \in \mathcal{U}_0(x, \pi_0)} \pi(a|x)q(x, a)\right], \tag{3}$$

where $\mathcal{U}_0(x, \pi_0) := \{a \in \mathcal{A} \mid \pi_0(a|x) = 0\}$ is the set of *deficient* actions for context $x$ under $\pi_0$. This bias is due to the fact that IPS cannot evaluate the actions that are not explored by the logging policy. To deal with this bias unavoidable for IPS, Sachdeva et al. (2020) suggests using a reward regression model $\hat{q}(x, a) \approx q(x, a)$ in the form of Doubly Robust (DR):

$$\hat{V}_{\text{DR}}(\pi; \mathcal{D}, \hat{q}) := \frac{1}{n}\sum_{i=1}^{n}\{w(x_i, a_i)(r_i - \hat{q}(x_i, a_i)) + \mathbb{E}_{\pi(a|x_i)}[\hat{q}(x_i, a)]\}. \tag{4}$$

While the bias caused by support deficiency and the variance can be smaller if $\hat{q}(x, a)$ is accurate (Dudík et al., 2014; Sachdeva et al., 2020), it is almost impossible to achieve such accuracy when only a small (like few-shot) dataset is available and there are new actions in the target domain.

## 3 CROSS-DOMAIN OFF-POLICY EVALUATION AND LEARNING

To achieve yet effective OPE/L even with no exploration and few-shot data in the target environment, this section proposes a new OPE problem that aims to estimate the value of a new policy, but we can have access not only to the historical data collected from the domain in which the new policy will be implemented (*target* domain), but also the datasets from other domains (*source* domains) that have varying data generation processes (DGPs). We also develop an estimator to effectively leverage logged data from both the target and source domains and extend it to a policy gradient method.

Here, we introduce $K$ different domains $\{k\}_{k=1}^{K}$ where the logged dataset is collected under potentially different logging policies in each domain. One out of $K$ domains is the target domain $T$, where we are interested in deploying a new policy, and the rest are source domains $S$. The DGP within each domain $k$ can be formulated as a respective contextual bandit process, i.e., we first observe the context $x^k \in \mathcal{X}$ from an unknown $p^k(x)$. Given context $x^k$, logging policy $\pi_0^k(a|x)$ chooses action $a^k$, and then the reward $r^k$ is observed from an unknown $p^k(r|x, a)$. Following this process, we observe logged bandit data denoted by $\mathcal{D}^k := \{(x_i^k, a_i^k, r_i^k)\}_{i=1}^{n^k} \sim p(\mathcal{D}^k) = \prod_{i=1}^{n_k} p^k(x_i)\pi_0^k(a_i|x_i)p^k(r_i|x_i, a_i)$ for each domain, where $n^k$ is the sample size of domain $k$. Note that we sometimes use $\mathcal{D} := \bigcup_{k=1}^{K}\mathcal{D}^k$ to denote available logged data across all domains.

We are now interested in evaluating a new policy $\pi(a|x)$ in the target domain whose policy value is defined using the target distributions, $(p^T(x), p^T(r|x, a))$, as follows.

$$V^T(\pi) := \mathbb{E}_{p^T(x)\pi(a|x)p^T(r|x,a)}[r] = \mathbb{E}_{p^T(x)\pi(a|x)}[q^T(x, a)], \tag{5}$$

where $q^T(x, a) := \mathbb{E}_{p^T(r|x,a)}[r|x, a]$ denotes the expected reward in the target domain given context $x$ and action $a$. Note here that we can regard the conventional formulation described in Section 2 as a special case where there exists only the target domain. Thus, IPS and DR can readily be defined in our formulation by using only the target domain data $\mathcal{D}^T$ as $\hat{V}_{\text{IPS}}(\pi; \mathcal{D}^T)$ and $\hat{V}_{\text{DR}}(\pi; \mathcal{D}^T, \hat{q}^T)$.

In contrast, a possible approach to perform the existing estimators with source domain data is to naively integrate the datasets from all domains when performing IPS or DR as below.

$$\hat{V}_{\text{IPS-ALL}}(\pi; \mathcal{D}) := \frac{1}{N}\sum_{k=1}^{K}\sum_{i=1}^{n^k} w^k(x_i^k, a_i^k)r_i^k, \tag{6}$$

$$\hat{V}_{\text{DR-ALL}}(\pi; \mathcal{D}) := \frac{1}{N}\sum_{k=1}^{K}\sum_{i=1}^{n^k}\left\{w^k(x_i^k, a_i^k)(r_i^k - \hat{q}(x_i^k, a_i^k)) + \mathbb{E}_{\pi(a|x_i^k)}[\hat{q}(x_i^k, a)]\right\}, \tag{7}$$

where $w^k(x, a) := \pi(a|x)/\pi_0^k(a|x)$ and $N := \sum_{k=1}^K n^k$. IPS-ALL and DR-ALL use information explored in source domains to estimate the policy value in the target domain with smaller variance by using more data. However, they completely ignore the differences in DGPs across different domains, possibly introducing substantial bias as we will demonstrate in our experiments.

## 3.1 THE COPE ESTIMATOR

This section proposes a new estimator called COPE, which effectively integrates logged bandit data from target and source domains to estimate the value of new policies in the target domain even with less data and no exploration within the target domain. The key to deriving our estimator is how to pool information from source domains to deal with the aforementioned challenging scenarios while not introducing much bias by taking into account the differences in DGPs. We achieve this by the following decomposition of the expected reward function.

$$q^k(x, a) = \underbrace{g(x, a, \phi(k))}_{\text{domain-cluster effect}} + \underbrace{h(x, a, k)}_{\text{domain-specific effect}}, \tag{8}$$

where $\phi(k)$ is a function to cluster similar domains. We can, for example, obtain a clustering function by heuristically performing an off-the-shelf clustering algorithm based on the empirical average of the rewards of each domain ($\bar{r}^k := \sum_{i=1}^{n_k} r_i/n_k$) as the domain embedding, which generally works well in our experiments. The domain-cluster effect $g(x, a, \phi(k))$ in Eq. (8) is a factor of the expected reward function that domains in the same cluster have in common, while the domain-specific effect $h(x, a, k)$ is the effect not solely modeled by the domain-cluster effect and thus is dependent on each individual domain $k$. For example, in a medical problem, the cluster effect could capture the shared effect of medical treatment within similar hospitals, while the domain-specific effect models how effective an action is for a particular hospital compared to the average in the cluster. Based on this decomposition, we design a new estimator by applying different estimation strategies between the domain-cluster ($g$) and specific ($h$) effects. Specifically, our estimator unbiasedly estimates the domain-cluster effect by applying the *multiple importance weighting* technique (Owen, 2013; Agarwal et al., 2017) and the domain-specific effect by reward regression as

$$\hat{V}_{\text{COPE}}(\pi; \mathcal{D}^{\phi(T)}) := \frac{1}{n^{\phi(T)}} \sum_{k \in \phi(T)} \sum_{i=1}^{n^k} \frac{\pi(a_i^k|x_i^k)}{p^{\phi(T)}(a_i^k|x_i^k)} (r_i^k - \hat{q}^T(x_i^k, a_i^k))$$

$$+ \frac{1}{n^T} \sum_{i=1}^{n^T} \sum_{a^T \in \mathcal{A}} \pi(a^T|x_i^T) \hat{q}^T(x_i^T, a^T), \tag{9}$$

where $\mathcal{D}^{\phi(T)}$ represents the logged data of domains belonging to the same cluster as the target domain (i.e., $\phi(T)$, which we call the *target cluster*), and $n^{\phi(T)} := \sum_{k \in \phi(T)} n^k$ is the total sample size of logged data within the target cluster $\phi(T)$. It should also be noted that we use the joint distribution of context and action under the logging policy averaged within the target cluster $\phi(T)$ in the denominator of the first term of the COPE estimator, which is formally defined as

$$p^{\phi(T)}(a|x) := \frac{1}{n^{\phi(T)}} \sum_{k \in \phi(T)} n^k \frac{p^k(x)}{p^T(x)} \pi_0^k(a|x), \tag{10}$$

which results in what is called multiple importance weighting (Owen, 2013; Agarwal et al., 2017), i.e., $\pi(a^k|x^k)/p^{\phi(T)}(a^k|x^k)$ in COPE to deal with varying distributions in the target cluster. In our experiments, we estimate the weight based only on observable logged data using techniques from density ratio estimation (Sugiyama et al., 2012; Kanamori et al., 2012), as detailed in Appendix C.

Our estimator is expected to substantially outperform the naive application of existing estimators when there is only limited/no exploration and less data in the target domain. First, COPE has a lower bias than IPS and DR by estimating the value of deficient and new actions by using data from source domains that are in the same cluster as the target domain, i.e., $\phi(T)$. COPE also provides substantial variance reduction in the case of few-shot data in the target domain by transferring information from source domains, while IPS and DR use only the target domain data. Moreover, COPE has lower bias than IPS-ALL and DR-ALL because it does not naively integrate data from all domains as IPS-ALL

and DR-ALL, but rather applies multiple importance weighting within the data in the target cluster, with adjustments for the varying context and action distributions across different domains.

In the following, we analyze the statistical properties of COPE and discuss its intriguing interpretation as a strict generalization of existing estimators.

**Condition 3.1** (Common Cluster Support). The logging policy $\pi_0^T$ is said to have common cluster support for policy $\pi$ if $\pi(a|x) > 0 \implies p^{\phi(T)}(a|x) > 0$ for all $a \in \mathcal{A}$ and $x \in \mathcal{X}$.

Note that, when Condition 2.1 is true, Condition 3.1 is always true, indicating that Condition 3.1 is more relaxed. In particular, when the logging policy is deterministic or there are new actions in the target domain, Condition 2.1 is never satisfied, while Condition 3.1 can still be satisfied when there is some exploration in the target cluster $\phi(T)$ (not necessarily in the target domain).

Based on this milder support condition, we provide the bias of COPE as below.

**Theorem 3.1** (Bias of COPE). Under Condition 3.1, COPE has the following bias.

$$\mathbb{E}_{p^T(x)\pi(a|x)}\left[\left\{\left(\sum_{k\in\phi(T)}\underbrace{\frac{n^k}{n^{\phi(T)}}\frac{p^k(x)}{p^T(x)}\frac{\pi_0^k(a|x)}{p^{\phi(T)}(a|x)}}_{:=\eta(k)}\Delta_{q,\hat{q}}^k(x,a)\right) - \Delta_{q,\hat{q}}^T(x,a)\right\}\right], \qquad (11)$$

where $\Delta_{q,\hat{q}}^k(x,a) := q^k(x,a) - \hat{q}^k(x,a)$ is an estimation error of the regression model $\hat{q}$ given context $x$ and action $a$ for domain $k$. Note that $\sum_{k\in\phi(T)}\eta(k) = 1$. See Appendix B.2 for the proof.

Theorem 3.1 suggests that the bias is characterized by the difference between the weighted average of the regression error in the target cluster $(\sum_{k\in\phi(T)}\eta(k)\Delta_{q,\hat{q}}^k(x,a))$ and the error in the target domain $(\Delta_{q,\hat{q}}^T(x,a))$. Therefore, if $\Delta_{q,\hat{q}}^k(x,a) \approx \Delta_{q,\hat{q}}^T(x,a) \implies q^k(x,a) - q^T(x,a) \approx \hat{q}^k(x,a) - \hat{q}^T(x,a)$ holds for $\forall k \in \phi(T)$ (i.e., $\hat{q}^k(x,a)$ accurately preserves the relative reward differences of actions, $q^k(x,a) - q^T(x,a)$, within $\phi(T)$), the bias of COPE becomes small. This is because COPE already unbiasedly estimates the domain-cluster effect via its multiple importance weighting, and thus it is sufficient for the regression model to estimate only relative reward differences of actions between domains in $\phi(T)$ and the target domain $T$ to make the global estimate low-bias. More formally, Theorem 3.1 implies the unbiasedness of COPE under the *CPC* condition.

**Condition 3.2** (Conditional Pairwise Correctness; CPC). A regression model $\hat{q}^k(x,a)$ and domain clustering function $\phi(k)$ satisfy conditional pairwise correctness if the following holds true:

$$q^k(x,a) - q^T(x,a) = \hat{q}^k(x,a) - \hat{q}^T(x,a),$$

for all $x \in \mathcal{X}$, $a \in \mathcal{A}$, and $k$ s.t. $k \in \phi(T)$.

**Corollary 3.1.** Under Conditions 3.1 and 3.2, COPE is unbiased, i.e., $\mathbb{E}_{\mathcal{D}^{\phi(T)}}[\hat{V}_{\text{COPE}}(\pi; \mathcal{D}^{\phi(T)})] = V(\pi)$. See Appendix B.2 for the proof.

Condition 3.2 requires that the regression model $\hat{q}$ should only correctly preserve the relative reward difference $q^k(x,a) - q^T(x,a)$ between domains in $\phi(T)$ and the target domain $T$. Therefore, it does not necessitate an accurate estimate of the absolute value of the reward function, which is a general requirement for regression-based estimators such as the direct method (DM). Note that CPC needs only an accurate estimation of the relative reward difference within the target cluster, and thus it does not require anything about source domains outside the particular cluster.

It is important to clarify that we do not expect Condition 3.2 to hold in practice. It is simply a condition that helps to understand when COPE can be unbiased. This is why we first present Theorem 3.1, which characterizes the bias of COPE without assuming Condition 3.2. It is also important to note that in challenging cases, such as deterministic logging and the presence of new actions, existing estimators, including IPS and DR, produce much bias, but our estimator leverages source domain data and is much more robust to the bias arising due to those challenges, as we will demonstrate.

It is also worth mentioning that the size of the target cluster plays a crucial role in deciding the bias-variance tradeoff of COPE. When $|\phi(T)|$ is small, the bias of the estimator becomes small. This is because decreasing the cluster size makes CPC milder. In the extreme case where $|\phi(T)| = 1$, COPE becomes unbiased irrespective of the accuracy of the regression model because CPC requires

nothing in that case. In contrast, the variance of COPE is likely to increase with a small cluster because it leads to a higher variation of its importance weights.

Here, we also provide an intriguing interpretation of COPE as a spectrum of DR and DR-ALL, with the number of domains in the cluster $|\phi(T)|$ being a lever of the tradeoff. More specifically, in the extreme case with $|\phi(T)| = 1$, meaning that there is only the target domain in the cluster, COPE reduces to DR because the averaged joint distribution $p^{\phi(T)}(a|x)$ reduces to the logging policy of the target domain, i.e., $\pi_0^T(a|x)$. On the other hand, when $|\phi(T)| = K$, meaning that the cluster $\phi(T)$ contains all $K$ domains, COPE reduces to DR-ALL. As a strict generalization of different versions of DR, COPE never performs worse than these existing methods with a good (if not perfect) tuning of the cluster size $|\phi(T)|$. Note that the cluster size $|\phi(T)|$ is a hyperparameter of COPE, and we can tune it based only on available logged data by using recent advancements in the relevant literature (Su et al., 2020b; Udagawa et al., 2023; Felicioni et al., 2024; Cief et al., 2024b).

## 3.2 EXTENSION TO CROSS-DOMAIN OFF-POLICY LEARNING

In addition to the OPE counterpart, we can formulate the problem of learning a new policy to optimize the expected reward in the target domain as $\max_\theta V^T(\pi_\theta)$ where $\theta \in \mathbb{R}^d$ is the policy parameter. A typical approach to solving the policy learning problem is the policy-based approach, which updates the policy parameter by iterative gradient ascent as $\theta_{t+1} \leftarrow \theta_t + \nabla_\theta V^T(\pi_\theta)$. Since the true gradient $\nabla_\theta V^T(\pi_\theta) = \mathbb{E}_{p^T(x)\pi_\theta(a|x)}[q^T(x,a)\nabla_\theta \log \pi_\theta(a|x)]$ is unknown, we need to estimate it from observable data, which has been done by applying IPS or DR in the existing literature (Su et al., 2019; Metelli et al., 2021). However, similarly to the case with OPE, under the violation of common support with deterministic logging and new actions, policy gradient estimators based on IPS and DR produce bias. Moreover, in the presence of new actions that become newly available when learning a new policy for the target domain, these conventional methods cannot evaluate and choose such new actions at all, even though some of them might have high expected rewards.

To solve these seemingly intractable problems in OPL, we can indeed readily extend COPE as a policy-gradient estimator to learn a new policy that optimizes the policy value of the target domain.

$$\nabla_\theta \hat{V}_{\text{COPE-PG}}^T(\pi_\theta; \mathcal{D}^{\phi(T)}) := \frac{1}{n^{\phi(T)}} \sum_{k \in \phi(T)} \sum_{i=1}^{n^k} \frac{\pi_\theta(a_i^k|x_i^k)}{p^{\phi(T)}(a_i^k|x_i^k)}(r_i^k - \hat{q}(x_i^k, a_i^k))\nabla_\theta \log \pi_\theta(a_i^k|x_i^k)$$

$$+ \frac{1}{n^T}\sum_{i=1}^{n^T}\mathbb{E}_{\pi_\theta(a^T|x_i^T)}[\hat{q}(x_i^T, a^T)\nabla_\theta \log \pi_\theta(a^T|x_i^T)] \tag{12}$$

As already discussed, our policy-gradient estimator in Eq. (12) particularly uses data from source domains that belong to the target cluster through multiple importance weighting, enabling to even learn the value of actions with little or no previous exploration.

## 4 EMPIRICAL ANALYSIS

**Dataset.** This section empirically demonstrates the advantages of COPE and COPE-PG against existing ideas on a real-world public dataset called KuaiRec (Gao et al., 2022) collected from a recommendation system on a video-sharing app. The small matrix of the dataset consists of 1,411 users (denoted as $u \in \mathcal{U}$), 3,327 items, and 4,676,570 interactions, with a density of 99.6%, which enables OPE/L experiments without synthetic reward functions. We randomly select 30 actions that have at least one interaction with all users for our experiments. We use the user features and watch ratio recorded in the original data as the context $x_u$ and expected reward $q(x_u, a)$, respectively.

**Setup.** To simulate our problem of cross-domain OPE/L, we sample users from a domain-specific distribution $p^k(u)$ for each domain $k$, which is defined as $p^k(u) := \frac{\exp(\alpha^k \cdot \bar{q}(u))}{\sum_{u' \in \mathcal{U}} \exp(\alpha^k \cdot \bar{q}(u'))}$, where $\bar{q}(u) := (1/|\mathcal{A}|) \sum_{a \in \mathcal{A}} q(x_u, a)$ and $\alpha^k$ is a parameter that controls the user distribution in each domain. A domain with a large positive $\alpha^k$ has a higher density of users with a larger $\bar{q}(u)$. We use different values of $\alpha^k$ for different domains to vary the context distributions across domains.

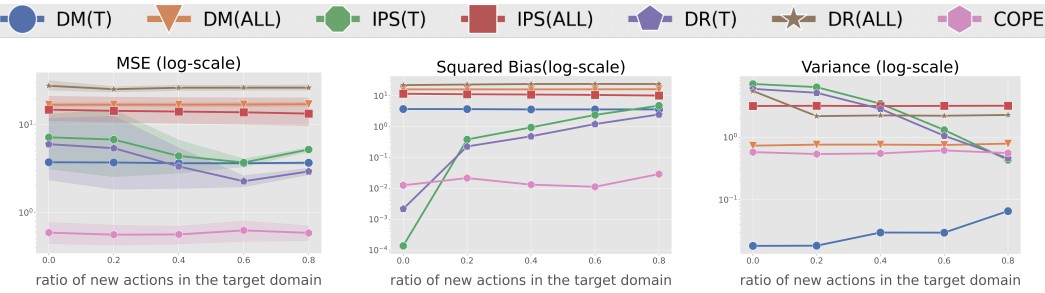

Figure 2: MSE(**left**), Squared Bias(**center**), and Variance(**right**) with varying ratios of new actions in the target domain.

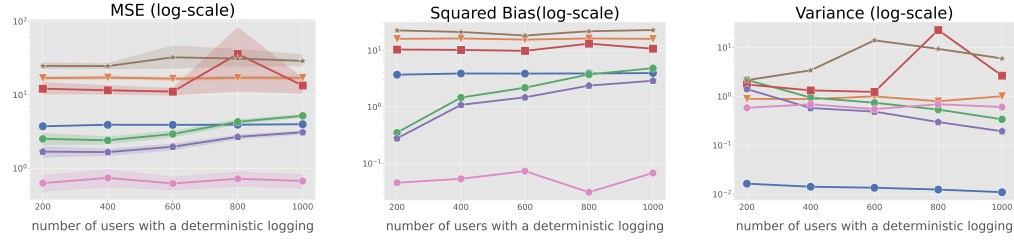

Figure 3: MSE(**left**), Squared Bias(**center**), and Variance(**right**) with varying numbers of users whose logging policy is deterministic in the target domain.

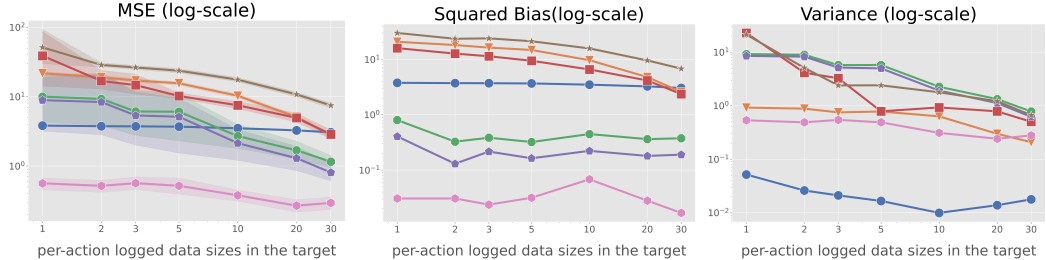

Figure 4: MSE(**left**), Squared Bias(**center**), and Variance(**right**) with varying (per-action) logged data sizes in the target domain.

We sample an action $a^k$ based on the domain-specific logging policy $\pi_0^k$, which is defined as below.

$$\pi_0^k(a \mid x_u) := \frac{\exp(\beta^k \cdot (q(x_u, a) + \eta_{u,a}))}{\sum_{a' \in \mathcal{A}} \exp(\beta^k \cdot (q(x_u, a') + \eta_{u,a'}))}, \tag{13}$$

where $\beta^k$ and $\eta_{u,a}$ are sampled from a uniform distribution within range $[-2.0, 2.0]$ and $[-3.0, 3.0]$, respectively. In contrast, the new policy $\pi$ is defined via the epsilon-greedy rule as

$$\pi(a \mid x) = (1 - \epsilon) \cdot \mathbb{I}\{a = \mathrm{argmax}_{a' \in A} \, q(x_u, a')\} + \epsilon / |\mathcal{A}|, \tag{14}$$

where $\epsilon \in [0, 1]$ controls the quality of $\pi$ and we set $\epsilon = 0.2$ as default. We sample the reward $r^k$ from a normal distribution with mean $q(x_u, a)$ and standard deviation $\sigma = 1$. Iterating this procedure $n^k$ times in each domain generate $\mathcal{D}^k$ with $n^k$ independent copies of $(u^k, x_u^k, a^k, r^k)$.

**Baselines.** We compare COPE with IPS(T), IPS-ALL, DR(T), and DR-ALL where IPS(T) and DR(T) perform off-policy estimation using only the target domain data $\mathcal{D}^T$. We also include the Direct Method (DM) ($\hat{V}_{\mathrm{DM}}(\pi; \mathcal{D}^T, \hat{q}^T)$) and DM-ALL ($\hat{V}_{\mathrm{DM-ALL}}(\pi; \mathcal{D}, \hat{q})$) as baselines, which are rigorously defined in the appendix. We use Random Forest (Breiman, 2001) implemented in scikit-learn (Pedregosa et al., 2011) along with 3-fold cross-fitting (Newey & Robins, 2018) to obtain $\hat{q}^T(x, a)$ for DR and DM, and $\hat{q}(x, a)$ for DR-ALL, DM-ALL, and COPE. In addition, for COPE, we use $|\phi(T)| = 4$, where we define the target cluster $\phi(T)$ by the set of domains for which the difference in the empirical average of the rewards, $|\bar{r}^k - \bar{r}^T|$, is small.

**Results: *Cross-Domain OPE*.** The following reports and discusses the MSE, squared bias, and variance of the OPE estimators computed over 200 sets of logged data, each replicated with different seeds. The shaded regions in the MSE plots represent the 95% confidence intervals estimated via bootstrap. Note that we set $K = 10$ for the number of domains and $n^k = 100$ for the logged data size of each domain as the default experimental parameters.

First, we compare the estimators under varying ratios of new actions in the target domain, $|\mathcal{U}_0^T|/|\mathcal{A}| \in \{0, 0.2, 0.4, 0.6, 0.8\}$, where $\mathcal{U}_0^T := \{a \in \mathcal{A} \,|\, \pi_0^T(a|x) = 0, \forall x\}$ in Figure 2. The results indicate that estimators using only logged data from the target domain, particularly IPS(T) and DR(T), produce larger bias as the ratio increases, while estimators that naively use logged data from all domains, such as IPS-ALL, DR-ALL, and DM-ALL, exhibit consistently high bias due to their inability to deal with differences in DGPs. In contrast, COPE consistently performs the best without being affected by the presence of new actions and producing much bias. Note that the MSEs of IPS and DR do not deteriorate rapidly as the number of new actions increases. This is because, although the bias of IPS and DR increases with new actions, their variance decreases as the number of supported actions decreases (a similar phenomenon is observed in Saito & Joachims (2022)).

Next, we compare the estimators under varying numbers of users whose logging policy is deterministic. We applied a logging policy that deterministically selects a single action to a subset of the users, while we apply a stochastic logging policy in Eq. (13) to the rest. Figure 3 shows that COPE is robust against an increasing number of users having a deterministic logging policy and maintains a small bias even when a deterministic policy is applied to about 70% ($\simeq 1000/|\mathcal{U}|$) of all users. On the other hand, IPS(T) and DR(T) exhibit an increasing bias as the number of users with a deterministic logging policy grows, which is consistent with Eq. (3). This is similar to the pattern observed with the increasing ratio of new actions and demonstrates the larger bias reduction by COPE in harder scenarios with more new actions and users with deterministic logging.

We also compare the estimators under varying per-action data sizes ($n^T/|\mathcal{A}|$) in Figure 4. We can see that COPE achieves the most accurate estimation, particularly with smaller logged data sizes in the target data. Specifically, when there is only a single data point per action, i.e., $n^T/|\mathcal{A}| = 1$, COPE outperforms the best baseline (DM) by a substantial margin ($\frac{\text{MSE}(\hat{V}_{\text{DM(T)}})}{\text{MSE}(\hat{V}_{\text{COPE}})} = 6.79$). Note also that the best baselines are different for different $n^T/|\mathcal{A}|$, while COPE generally performs the best in any per-action data size. This powerful and stable behavior of COPE is due to its variance reduction by using data from multiple domains while accounting for varying DGPs to avoid producing bias.

**Results: *Cross-Domain OPL*.** We now compare the estimators in terms of their effectiveness in OPL when used to estimate the policy gradient $\nabla_\theta V^T(\pi_\theta)$. In the following, we report the target policy values relative to those of the logging policy, computed over 350 sets of logged data, each replicated with different seeds. The default configurations are the same as those in the OPE experiments.

We first compare OPL methods under varying ratios of new actions in the target domain in Figure 5 (left). The results indicate that COPE-PG generally outperforms other methods without being affected by the presence of new actions. Additionally, Figure 5 (center) illustrates the policy value within new actions, $\mathbb{E}_{p^T(x)}[\sum_{a \in \mathcal{U}_0^T} \pi_\theta(a|x)q(x,a)]$ (Figure 5 (right) reports the normalized version: $\mathbb{E}_{p^T(x)}[\sum_{a \in \mathcal{U}_0^T} \pi_\theta(a|x)q(x,a)/\sum_{a \in \mathcal{U}_0^T} \pi_\theta(a|x)]$). We observe that methods using only logged data from the target domain, such as Reg-based(T), IPS-PG(T), and DR-PG(T), are not able to gain policy values from new actions, which is reasonable. In contrast, methods that additionally leverage logged data from the source domains, such as IPS-ALL, DR-ALL, and COPE-PG, develop policies that select new actions relatively well, with COPE-PG achieving particularly efficient selection of new actions, even though there is no exploration of such actions in the target domain.

We then compare the OPL methods under varying numbers of users with a deterministic logging policy in Figure 6 (left). We observed that the policy value of IPS-PG(T) and DR-PG(T) deteriorates as the degree of determinism increases. This is due to the fact that under deterministic logging policies and severe common support violations, they become substantially biased. On the other hand, COPE-PG consistently maintains the best policy value, unaffected by the increasing number of users with a deterministic logging policy due to its milder support condition (Condition 3.1).

We also compare the OPL methods under varying per-action training data sizes ($n^T/|\mathcal{A}|$) in the target domain in Figure 6 (center). As we can observe from the figure, most existing methods show that the policy value becomes higher as the training data size increases, but COPE-PG generally

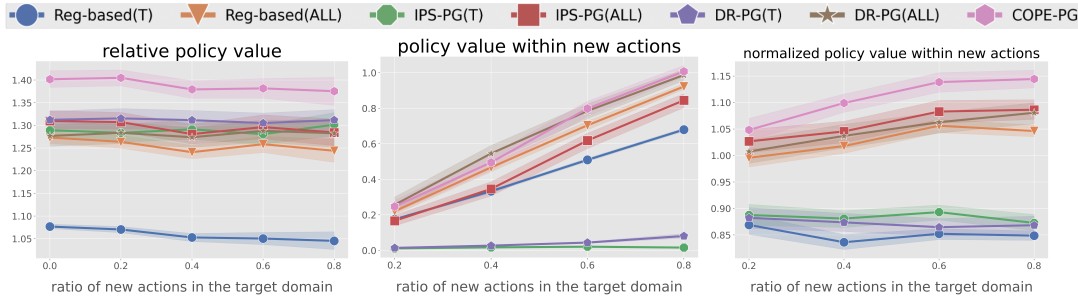

Figure 5: Comparison of the (**left**) the test policy values $V^T(\pi)$ (normalized by $V(\pi_0)$), (**center**) the test policy values within new actions, (**rigt**) the normalized test policy values within new actions, under varying ratios of new actions in the target domain.

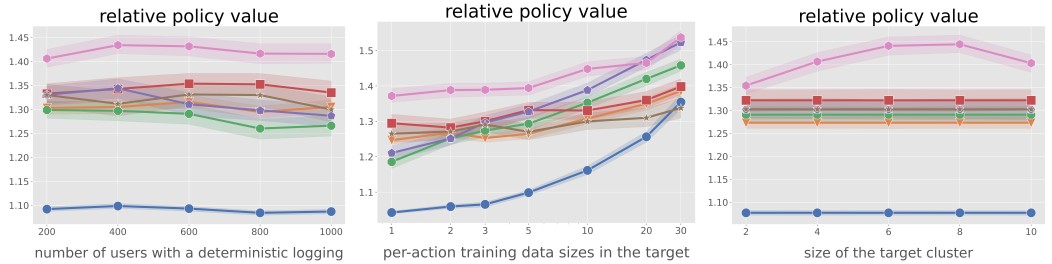

Figure 6: Comparison of the test policy values $V^T(\pi)$ (normalized by $V(\pi_0)$) of the OPL methods under (**left**) varying numbers of users whose logging is deterministic, (**center**) varying (per-action) training data sizes in the target domain, (**right**) varying sizes of the target cluster.

performs the best. It is particularly impressive to see the stable performance of COPE-PG, while no baseline methods generally perform well for varying training data sizes. Specifically, IPS-PG(ALL) and DR-PG(ALL) are the second best, following COPE-PG when $n^T/|\mathcal{A}| = 1$, while they perform even worse than IPS-PG(T) and DR-PG(T) when $n^T/|\mathcal{A}| = 30$. In contrast, IPS-PG(T) and DR-PG(T) perform similarly to COPE-PG when $n^T/|\mathcal{A}| = 30$ while they perform substantially worse when $n^T/|\mathcal{A}| = 1$. Given the stable performance of COPE-PG, there is no particular reason to prioritize other methods over ours in any $n^T/|\mathcal{A}|$ in terms of the OPL effectiveness.

We finally evaluate the effectiveness of COPE-PG with varying sizes of the target cluster $|\phi(T)|$ (a key hyperparameter of COPE-PG) in Figure 6 (right). Note that the baseline methods are independent of this parameter, so their results remain flat. The figure shows that the size of the target cluster indeed affects the performance of COPE-PG, with the best performance observed at moderate cluster sizes, such as 6 and 8. However, it should also be noted that COPE-PG still outperforms the baseline methods even with overly small ($|\phi(T)| = 2$) or large ($|\phi(T)| = 10$) cluster sizes, demonstrating the robustness of our method to potential failures in tuning of the parameter.

## 5 CONCLUSION AND FUTURE WORK

This paper studied OPE/L in challenging scenarios such as with deterministic logging policies, new actions, and few-shot logged data, which existing formulations and methods cannot address. To solve those challenging issues, we formulated the problem of cross-domain OPE/L and proposed a novel estimator and policy-gradient method based on a reward function decomposition to leverage data explored in source domains without introducing much bias. In particular, our methods are able to evaluate and optimize new policies even with deterministic logging policies or new actions in the target domain with analyzable bias.

As future work, even though the heuristic domain clustering of using the empirical averaged rewards as domain embeddings worked satisfactorily in our experiments, it would be valuable to develop a more principled method to perform clustering of domains where we can possibly apply techniques from relevant work in OPE (Peng et al., 2023; Sachdeva et al., 2024; Kiyohara et al., 2024b).

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

# A    RELATED WORK

Off-policy evaluation (OPE) and learning (OPL) have gained particular attention in contextual bandit settings as they offer a safe and cost-efficient alternative to online A/B testing (Mehrotra et al., 2018; Gilotte et al., 2018; Saito et al., 2021; Kiyohara et al., 2024a) and are of high practical relevance. While several practical estimators and policy gradient methods (most of which are based on importance weighting) already exist (Saito & Joachims, 2021; Uehara et al., 2022; Dudík et al., 2014; Wang et al., 2017; Su et al., 2020a) for each of OPE or OPL, the effectiveness of these methods can decline in challenging but realistic scenarios. First, high bias may occur when the logging policy fails to satisfy the common support condition (Sachdeva et al., 2020), which becomes even more severe when the logging policy is deterministic or when there are new actions. Several recent works have addressed violations of common support (also known as support deficiency) by exploiting additional information about actions, restricting the action space, applying reward extrapolation, or limiting the policy space, but they cannot handle deterministic logging policies or entirely new actions (Felicioni et al., 2022; Sachdeva et al., 2020). It may also be helpful to use some structure in the action or reward space to relax the common support requirement, as explored by Saito & Joachims (2022); Saito et al. (2023); Cief et al. (2024a); Taufiq et al. (2023); Sachdeva et al. (2024); Kiyohara et al. (2024b). For example, Saito & Joachims (2022) leverage additional information about the actions in the form of action embeddings, and Kiyohara et al. (2023) define importance weights in a low-dimensional slate abstraction space. However, this useful structure is not always learnable. In addition, existing estimators and methods based on importance weighting face significant variance issues, particularly when the logged dataset is small and the action space is large (Saito & Joachims, 2022; Cief et al., 2024a; Sachdeva et al., 2024). Weight clipping (Swaminathan & Joachims, 2015a; Su et al., 2020a) or normalization (Swaminathan & Joachims, 2015b) might be applied to mitigate variance, but they often introduce a substantial amount of bias in estimation.

Another line of work is OPE/L under covariate shift, as researched by Uehara et al. (2020). This research focuses on cases where the context distributions differ between the logged data ($p^{\text{hist}}(x)$) and the evaluation environment ($p^{\text{eval}}(x)$), while the reward distribution $p(r|x, a)$ remains unchanged. Under this setup, Uehara et al. (2020) propose doubly robust and efficient estimators by leveraging an estimator of the density ratio between the historical and evaluation data distributions. In contrast, we aim to estimate the same estimand using data from both target and source domains (which may include multiple source domains), with each domain having unique context distributions $p^k(x)$ and reward distributions $p^k(r|x, a)$. By leveraging this new setup, our primary goal is to address challenging scenarios in the target domain, such as new actions, deterministic logging policies, and extremely small logged data, which clearly differ from the motivations of Uehara et al. (2020).

Finally, we mention the work by Cief et al. (2024b) and Liu et al. (2024), which formulate the problem of cross-domain offline reinforcement learning (offline RL). The goals of their formulations are somewhat related to ours, as they aim to learn a policy that performs well in the target domain by leveraging data from both target and source domains. However, they focus on the offline RL setup and consider only shifts in transition dynamics. In contrast, we focus on the contextual bandit setup and address varying context and reward distributions across domains. We also analyze and experiment with the problem of OPE, while Cief et al. (2024b) and Liu et al. (2024) focus solely on policy learning. Furthermore, it is important to note that we explore the problem of cross-domain OPE/OPL to tackle the unsolved issues of new actions, deterministic logging, and limited logged data, whereas previous work more generally considers the cross-domain setup. Note that we did not compare pessimistic OPL methods (Swaminathan & Joachims, 2015a; London & Sandler, 2019; Gabbianelli et al., 2024; Sakhi et al., 2024) because they are not relevant to our context. Our main motivation is to solve the prevalent problem of (completely) deterministic logging and new actions, issues that pessimistic techniques do not aim to address. However, our proposed method could easily be combined with a pessimistic approach if one wants to do so.

# B  OMITTED PROOFS

## B.1  PROOF OF THEOREM 3.1

*Proof.* The bias of COPE under Condition 3.1 is derived as

$$
\mathrm{Bias}(\hat{V}_{\mathrm{COPE}}(\pi; D^{\phi(T)}))
$$
$$
= \mathbb{E}[\hat{V}_{\mathrm{COPE}}(\pi; D^{\phi(T)})] - V^T(\pi)
$$
$$
= \frac{1}{n^{\phi(T)}} \sum_{k \in \phi(T)} \sum_{i=1}^{n^k} \mathbb{E}_{p^k(x)\pi_0^k(a|x)p^k(r|x,a)}\Big[\frac{\pi(a_i^k|x_i^k)}{p^{\phi(T)}(a_i^k|x_i^k)}(r_i^k - \hat{q}(x_i^k, a_i^k))\Big]
$$
$$
+ \frac{1}{n^T} \sum_{i=1}^{n^T} \mathbb{E}_{p^T(x)\pi_0^T(a|x)p^T(r|x,a)}\big[\mathbb{E}_{\pi(a|x_i)}\hat{q}(x_i^T, a^T)\big]
$$
$$
- \mathbb{E}_{p^T(x)\pi(a|x)p^T(r|x,a)}[r]
$$
$$
= \sum_{k \in \phi(T)} \frac{n^k}{n^{\phi(T)}} \mathbb{E}_{p^k(x)\pi_0^k(a|x)p^k(r|x,a)}\Big[\frac{\pi(a^k|x^k)}{p^{\phi(T)}(a^k|x^k)}(r^k - \hat{q}(x^k, a^k))\Big]
$$
$$
+ \mathbb{E}_{p^T(x)\pi_0^T(a|x)p^T(r|x,a)}\Big[\sum_{a \in \mathcal{A}} \pi(a|x^T)\hat{q}(x^T, a)\Big]
$$
$$
- \mathbb{E}_{p^T(x)\pi(a|x)}[q^T(x, a)]
$$
$$
= \sum_{k \in \phi(T)} \frac{n^k}{n^{\phi(T)}} \mathbb{E}_{p^k(x)\pi_0^k(a|x)}\Big[\frac{\pi(a^k|x^k)}{p^{\phi(T)}(a^k|x^k)}\Delta_{q,\hat{q}}^k(x, a)\Big]
$$
$$
+ \sum_{x \in \mathcal{X}} \sum_{a \in \mathcal{A}} p^T(x)\pi(a|x)\hat{q}(x, a)
$$
$$
- \sum_{x \in \mathcal{X}} \sum_{a \in \mathcal{A}} p^T(x)\pi(a|x)q^T(x, a)
$$
$$
= \sum_{x \in \mathcal{X}} \sum_{k \in \phi(T)} \sum_{a \in \mathcal{A}} \frac{p^T(x)\pi(a|x)}{\sum_{k \in \phi(T)} \frac{n^k}{n^{\phi(T)}} p^k(x)\pi_0^k(a|x)} \frac{n^k}{n^{\phi(T)}} p^k(x)\pi_0^k(a|x)\Delta_{q,\hat{q}}^k(x, a)
$$
$$
- \sum_{x \in \mathcal{X}} \sum_{a \in \mathcal{A}} p^T(x)\pi(a|x)\Delta_{q,\hat{q}}^T(x, a)
$$
$$
= \sum_{x \in \mathcal{X}} p^T(x)\Big\{\Big(\sum_{k \in \phi(T)} \sum_{a \in \mathcal{A}} \frac{\pi(a|x)}{\sum_{k \in \phi(T)} \frac{n^k}{n^{\phi(T)}} p^k(x)\pi_0^k(a|x)} \frac{n^k}{n^{\phi(T)}} p^k(x)\pi_0^k(a|x)\Delta_{q,\hat{q}}^k(x, a)\Big)
$$
$$
- \sum_{a \in \mathcal{A}} \pi(a|x)\Delta_{q,\hat{q}}^T(x, a)\Big\}
$$
$$
= \sum_{x \in \mathcal{X}} p^T(x) \sum_{a \in \mathcal{A}} \pi(a|x)\Big\{\Big(\sum_{k \in \phi(T)} \frac{1}{\sum_{k \in \phi(T)} \frac{n^k}{n^{\phi(T)}} p^k(x)\pi_0^k(a|x)} \frac{n^k}{n^{\phi(T)}} p^k(x)\pi_0^k(a|x)\Delta_{q,\hat{q}}^k(x, a)\Big)
$$
$$
- \Delta_{q,\hat{q}}^T(x, a)\Big\}
$$
$$
= \mathbb{E}_{p^T(x)\pi(a|x)}\left[\left\{\left(\sum_{k \in \phi(T)} \frac{n^k}{n^{\phi(T)}} \frac{p^k(x)}{p^T(x)} \frac{\pi_0^k(a|x)}{p^{\phi(T)}(a|x)}\Delta_{q,\hat{q}}^k(x, a)\right) - \Delta_{q,\hat{q}}^T(x, a)\right\}\right].
$$

□

### B.2 PROOF OF COROLLARY 3.1

*Proof.* Starting from the final expression of B.1, the bias of COPE under the additional Condition 3.2 is calculated as

$$\mathbb{E}_{p^T(x)\pi(a|x)}\left[\left\{\left(\sum_{k\in\phi(T)}\frac{n^k}{n^{\phi(T)}}\frac{p^k(x)}{p^T(x)}\frac{\pi_0^k(a|x)}{p^{\phi(T)}(a|x)}\Delta_{q,\hat{q}}^k(x,a)\right)-\Delta_{q,\hat{q}}^T(x,a)\right\}\right]$$

$$=\mathbb{E}_{p^T(x)\pi(a|x)}\left[\left\{\left(\sum_{k\in\phi(T)}\frac{n^k}{n^{\phi(T)}}\frac{p^k(x)}{p^T(x)}\frac{\pi_0^k(a|x)}{p^{\phi(T)}(a|x)}\Delta_{q,\hat{q}}^T(x,a)\right)-\Delta_{q,\hat{q}}^T(x,a)\right\}\right]\because\text{Condition 3.2}$$

$$=\mathbb{E}_{p^T(x)\pi(a|x)}\left[\Delta_{q,\hat{q}}^T(x,a)\left\{\left(\sum_{k\in\phi(T)}\frac{n^k}{n^{\phi(T)}}\frac{p^k(x)}{p^T(x)}\frac{\pi_0^k(a|x)}{p^{\phi(T)}(a|x)}\right)-1\right\}\right]$$

$$=0,$$

which concludes the proof. $\square$

### B.3 BIAS OF THE BASELINE METHODS

Here, we can derive the bias of IPS-ALL by calculating its expectation as

$$\mathbb{E}_{\mathcal{D}}[\hat{V}_{\text{IPS-ALL}}(\pi;\mathcal{D})]=\frac{1}{N}\sum_{k=1}^{K}\sum_{i=1}^{n^k}\mathbb{E}_{p^k(x)\pi_0^k(a|x)}[w^k(x^k,a^k)q^k(x,a)]$$

$$=\frac{1}{N}\sum_{k=1}^{K}\sum_{i=1}^{n^k}\mathbb{E}_{p^k(x)\pi^k(a|x)}[q^k(x,a)]\because w^k(x^k,a^k)=\frac{\pi^k(a\,|\,x)}{\pi_0^k(a\,|\,x)}$$

$$=\sum_{k=1}^{K}\frac{n^k}{N}\mathbb{E}_{p_k(x)\pi(a|x)}[q^k(x,a)].$$

Therefore, the bias of IPS-ALL is:

$$\text{Bias}(\hat{V}_{\text{IPS}}(\pi;\mathcal{D}))=\left|\mathbb{E}_{p^T(x)\pi(a|x)}[q^T(x,a)]-\left(\sum_{k}\frac{n^k}{N}\mathbb{E}_{p_k(x)\pi(a|x)}[q^k(x,a)]\right)\right|.$$

We observe that they are no longer guaranteed to remain unbiased in our setup. This bias becomes significant when the reward distributions across domains differ substantially because the expectation of IPS-ALL is characterized by the weighted sum of the policy values of all the domains involved. In contrast, the bias becomes zero when the reward distributions across all domains are identical (which is no longer a cross-domain setup). Based on this derivation and Theorem 3.1, we can compare the bias of COPE and that of IPS/DR-ALL as follows:

- COPE performs better when the distributions across domains are substantially different and also when the pairwise estimation by $\hat{q}(x,a)$ is accurate.
- IPS/DR-ALL performs better when the distributions across domains are similar, i.e., when the cross-domain setup is close to a typical single-domain setup.

When the setup is close to the typical single-domain scenario, effective estimators such as DR and its extensions are already well-known. Therefore, we focused more on novel and challenging scenarios where the distributions across domains are not necessarily similar, as baseline methods tend to fail in these cases as shown above and in our experiments.

In practice, we do not precisely know which of COPE and baselines are better. This is not the case only for our setup, but also for the other OPE problems. A typical example is that we never know

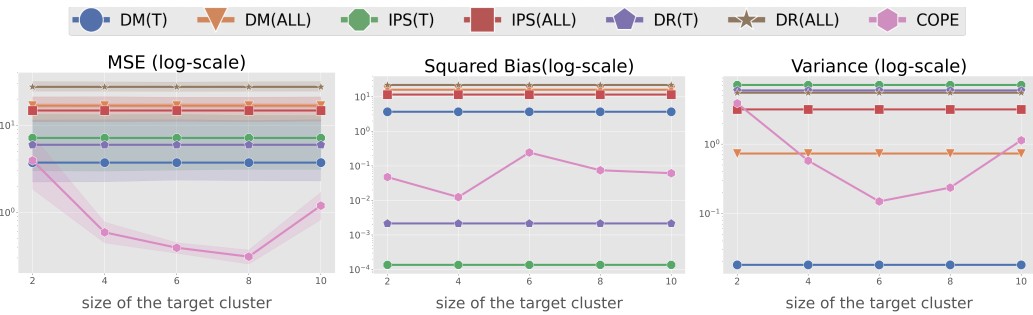

Figure 7: MSE(**left**), Squared Bias(**center**), and Variance(**right**) with with varying size of the target cluster.

which of IPS and DM is better, because the comparison depends on many (potentially unknown) parameters such as the reward noise and logged data size. This is why there exists an orthogonal line of work around estimator selection in OPE (Su et al., 2020b; Udagawa et al., 2023; Cief et al., 2024b), and we can use those methods to identify the appropriate estimator tailored to the specific problem when we use OPE in general.

## C  DENSITY RATIO ESTIMATION

Our implementation in the experiments relies on one of the most standard methods, unconstrained Least-Squares Importance Fitting (uLSIF), to perform density ratio estimation proposed in (Kanamori et al., 2012; Sugiyama et al., 2012). The method considers minimizing the following squared error between $s$ and $r$ where $s$ is an estimator for the ratio and $w(x) = p^k(x)/p^T(x)$ is the ratio we need to estimate in order to use $p^{\phi(T)}(x, a)$ in Eq. (10):

$$\mathbb{E}[(s(x) - w(x))^2] = \mathbb{E}_{p^k(x)}[(w(x))^2] - 2\mathbb{E}_{p^T(x)}[s(x)] + \mathbb{E}_{p^k(x)}[(s(x))^2]$$

The first term of the equation does not affect the result of the optimization and we can ignore it, i.e., the density ratio is estimated through the following minimization problem:

$$s^* = \arg\min_{s \in \mathcal{S}} \left[ \frac{1}{2}\mathbb{E}_{p^T(x)}[(s(x))^2] - \mathbb{E}_{p^k(x)}[s(x)] \right]$$

To solve the empirical version of this optimization, Kanamori et al. (2012) use kernel-based hypotheses to estimate the density ratio nonparametrically with a regularization term as usual. https://github.com/hoxo-m/densratio_py, which we relied on in our experiments, is one of the well-known public implementations of the method.

## D  ADDITIONAL EXPERIMENT SETUP AND RESULTS

### D.1  ADDITIONAL RESULTS ON REAL-WORLD DATA

This section compares the OPE estimators under varying sizes of the target cluster $|\phi(T)| \in \{2, 4, 6, 8, 10\}$ (a key hyperparameter of COPE) in Figure 7. Note that the baseline estimators are independent of this parameter, so their results remain constant. We observe that COPE, when using a smaller target cluster size, has a small bias but a large variance. This is because the smaller target cluster size reduces the bias caused by differences in the distribution across domains. In contrast, the smaller size also decreases the amount of available logged data, leading to an increase in variance. Additionally, the figure suggests that COPE with a larger target cluster size exhibits higher variance. This occurs because as the number of domains included in the COPE estimation increases, larger variances are more likely to arise from the density ratio terms. Nevertheless, the results show that COPE outperforms the baseline estimators, even with small ($|\phi(T)| = 2$) or large ($|\phi(T)| = 10$) cluster sizes, as seen in the OPL experiment.

Moreover, we exclude $\bar{q}(u)$ from the definition of the user distribution $p^k(u)$, following an observation that this might favor our methods. Instead, we define a linear function of the user feature $f^k(u) = \beta^k x_u$, where the coefficient vector $\beta^k$ is sampled from a normal distribution separately for each domain $k$. We then define the user distribution $p^k(u)$ for each domain as $p^k(u) := \frac{\exp(f^k(u))}{\sum_{u' \in \mathcal{U}} \exp(f^k(u'))}$. This procedure follows the multi-domain recommendation literature, which uses user features to define the domains in their experiments (Yang et al., 2024). More importantly, this approach avoids any dependence of the distribution on $\bar{q}(u)$. We conducted both OPE and OPL experiments based on this new user distribution, and the results are summarized in Tables 1 to 4. The results indicate that our method mostly outperforms the baseline methods that use only the target domain data or all the data across target and source domains in both OPE and OPL experiments (bold fonts indicate the best method, if the best and second best are not significantly different, we use bold fonts for both), even with $p^k(u)$ independent of $\bar{q}(u)$.

Furthermore, we can extend the definition of the user distribution as:

$$p^k(u) := \frac{\exp(\lambda \bar{q}(u) + (1 - \lambda) f^k(u))}{\sum_{u' \in \mathcal{U}} \exp(\lambda \bar{q}(u') + (1 - \lambda) f^k(u'))}$$

where $\lambda$ controls the degree to which the user distribution depends on $\bar{q}(u)$. When $\lambda = 1$, the setup reduces to our original experiment. Table 5 shows the result of the OPL experiment for varying $\lambda$ values. We should mention here that the difference between the proposed method and the baselines decreases when $\lambda$ is small, which is reasonable. These additional observations, however, demonstrate the robustness of COPE to varying user distributions, even with a simple clustering procedure. Nevertheless, we believe a more principled and effective approach to domain clustering for our method likely exists. As mentioned in Section 5, we consider the development of a clustering algorithm tailored to Cross-Domain OPE and OPL to be a valuable direction for future research.

### D.2 SYNTHETIC EXPERIMENT

This section empirically demonstrates the advantages of COPE over existing approaches using synthetic data. The benefits of using synthetic data include the ability to accurately compare estimators based on ground-truth policy values and the flexibility to control experiment configurations in order to test a wide range of scenarios. To generate the synthetic data, we first randomly cluster a total of $K = 30$ domains, corresponding to the definition of $\phi(k)$. We then sample a 5-dimensional domain embedding $e^k$ for each domain $k$ from the standard normal distribution. We also sample 10-dimensional context vectors $x^k$ from a normal distribution with mean $\mu_k$ and a standard deviation of $\sigma_x = 1$. The mean parameter $\mu_k$ of the context distribution is sampled from a uniform distribution within the range $[-1.0, 1.0]$.

We synthesize the expected reward function by using domain embedding $e$ as an input as below.

$$q(x, a, e; \lambda) = \lambda g(x, a, c) + h(x, a, e), \tag{15}$$

where $\lambda$ is an experiment parameter that controls the influence of the domain-cluster effect ($g$) compared to the domain-specific effect ($h$), where we use $\lambda = 0.5$ as the default parameter. Specifically, we used the following functions as $g(\cdot, \cdot, \cdot)$ (domain-cluster effect) and $h(\cdot, \cdot, \cdot)$ (domain-specific effect), respectively.

$$g(x, a, c) := x^\top M_{x,a}^c \text{one\_hot}_a + \theta_{x,c}^\top x + \theta_{a,c}^\top \text{one\_hot}_a + \theta_c^\top \text{one\_hot}_c,$$
$$h(x, a, e) := \theta_e^\top e + x^\top M_{x,e} e + \text{one\_hot}_a^\top M_{a,e} e,$$

where $\text{one\_hot}_a$ is the one-hot encoding of the action and $\text{one\_hot}_c$ is the one-hot encoding of the domain-cluster $c$. $M_{x,a}^c, \theta_{x,c}, \theta_{a,c}, \theta_c$ are parameter matrices or vectors in domain-cluster effect function sampled from a uniform distribution with range $[-1, 1]$ separately for each given domain cluster $c$. Also, $M_{x,e}, M_{a,e}$ are parameter matrices in domain-specific effect function sampled from a uniform distribution with range $[-1, 1]$, and $\theta_e$ is parameter vector sampled from a uniform distribution with range $[-10, 10]$ separately for each given domain cluster $c$.

Based on the expected reward function, we sample the reward $r^k$ from a normal distribution with mean $q(x, a, e; \lambda)$ and standard deviation $\sigma_r = 1$.

Table 1: MSE in OPE for varying ratios of new actions in the target domain

| ratios of new actions | 0.0 | 0.2 | 0.4 | 0.6 | 0.8 |
|---|---|---|---|---|---|
| **COPE(Ours)** | **1.1641** | **1.1647** | **0.8066** | **1.0663** | **1.0801** |
| DR(Target Domain) | 4.3283 | 4.2225 | 2.3985 | 2.9498 | 3.7964 |
| DR(ALL Domain) | 2.7582 | 2.2401 | 3.5354 | 3.3172 | 1.4530 |
| IPS(Target Domain) | 5.6442 | 4.9797 | 3.6800 | 4.8295 | 6.5302 |
| IPS(ALL Domain) | 2.1763 | 2.2538 | 2.2006 | 2.1881 | 2.9360 |

Table 2: MSE in OPE for varying numbers of users with deterministic logging in the target domain

| numbers of users with deterministic logging | 200 | 400 | 600 | 800 | 1000 |
|---|---|---|---|---|---|
| **COPE(Ours)** | **1.3238** | **1.3120** | **1.2926** | **1.2125** | **1.2172** |
| DR(Target Domain) | 2.3200 | 2.8398 | 3.2755 | 3.7992 | 4.4499 |
| DR(ALL Domain) | 2.5240 | 2.7915 | 3.0358 | 2.9843 | 3.3314 |
| IPS(Target Domain) | 3.2026 | 3.9641 | 4.6112 | 5.6947 | 7.0049 |
| IPS(ALL Domain) | 2.1561 | 2.1630 | 2.1906 | 2.2188 | 2.2076 |

Table 3: Test (Relative) Policy Value in OPL for varying ratios of new actions in the target domain

| ratios of new actions | 0.0 | 0.2 | 0.4 | 0.6 | 0.8 |
|---|---|---|---|---|---|
| **COPE-PG(Ours)** | **1.4463** | **1.4060** | **1.4389** | **1.3884** | 1.4090 |
| DR-PG(Target Domain) | 1.3518 | 1.3220 | 1.3364 | 1.2663 | 1.3339 |
| DR-PG(ALL Domain) | 1.4308 | 1.3947 | 1.4064 | 1.3420 | **1.4403** |
| IPS-PG(Target Domain) | 1.3119 | 1.2755 | 1.2899 | 1.2389 | 1.3350 |
| IPS-PG(ALL Domain) | 1.3588 | 1.3198 | 1.3267 | 1.3134 | 1.3246 |

Table 4: Test (Relative) Policy Value in OPL for varying numbers of users with deterministic logging in the target domain

| numbers of users with deterministic logging | 200 | 400 | 600 | 800 | 1000 |
|---|---|---|---|---|---|
| **COPE-PG(Ours)** | 1.4408 | **1.4531** | **1.4769** | **1.4669** | 1.4139 |
| DR-PG(ALL Domain) | **1.4477** | 1.4180 | 1.3954 | 1.4059 | **1.5115** |
| DR-PG(Target Domain) | 1.3101 | 1.3014 | 1.3091 | 1.3596 | 1.3013 |
| IPS-PG(ALL Domain) | 1.3381 | 1.3003 | 1.3149 | 1.3625 | 1.4651 |
| IPS-PG(Target Domain) | 1.2795 | 1.2770 | 1.2905 | 1.2765 | 1.2467 |

Table 5: Test (Relative) Policy Value in OPL for varying $\lambda$ values

| lambda values | 0.0 | 0.2 | 0.4 | 0.6 | 0.8 | 1.0 |
|---|---|---|---|---|---|---|
| **COPE-PG(Ours)** | **1.4713** | **1.4646** | **1.4739** | **1.4502** | **1.4780** | **1.4404** |
| DR-PG(ALL Domain) | 1.3812 | 1.3558 | 1.3475 | 1.3592 | 1.3042 | 1.3657 |
| DR-PG(Target Domain) | 1.4639 | 1.4334 | 1.3823 | 1.4374 | 1.3956 | 1.2799 |
| IPS-PG(ALL Domain) | 1.3553 | 1.3010 | 1.3133 | 1.2869 | 1.2770 | 1.2755 |
| IPS-PG(Target Domain) | 1.3629 | 1.3580 | 1.3572 | 1.3480 | 1.4390 | 1.3889 |

We define the logging policy $\pi_0^k$ for each domain $k$ by applying softmax to the expected reward function $q(x, a, e; \lambda)$ as

$$\pi_0^k(a|x) := \frac{\exp(\beta^k \cdot (q(x, a, e; \lambda) + \eta_{x,a})}{\sum_{a' \in \mathcal{A}} \exp(\beta^k \cdot (q(x, a', e; \lambda) + \eta_{x,a})} \qquad (16)$$

where $\beta^k$ and $\eta_{x,a}$ are sampled from a uniform distribution within range $[-0.5, 0.5]$ . In contrast, the evaluation policy $\pi$ is defined as

$$\pi(a|x) = (1 - \epsilon) \cdot \mathbb{I}\{a = \mathrm{argmax}_{a' \in A} \quad q(x, a', e; \lambda)\} + \frac{\epsilon}{|\mathcal{A}|}, \qquad (17)$$

where $\epsilon \in [0, 1]$ controls the quality of $\pi$ and we set $\epsilon = 0.2$ as default.

### D.2.1 BASELINE ESTIMATORS

Below, we define the baseline estimators compared in our (both synthetic and real-world) experiments.

**Direct Method (DM).** The DM estimator, which uses only logged data from the target domain, is defined as follows:

$$\hat{V}_{\mathrm{DM}}(\pi; \mathcal{D}^T, \hat{q}^T) := \frac{1}{n^T} \sum_{i=1}^{n^T} \mathbb{E}_{\pi(a|x_i)}[\hat{q}^T(x_i, a)],$$

where $\hat{q}^T(x, a)$ estimates $q^T(x, a)$ based on logged bandit data from the target domain. Additionally, the DM-ALL estimator, which uses logged data from both the target and source domains, is defined as follows:

$$\hat{V}_{\mathrm{DM-ALL}}(\pi; \mathcal{D}, \hat{q}) := \frac{1}{N} \sum_{k=1}^{K} \sum_{i=1}^{n^k} \mathbb{E}_{\pi(a|x_i^k)}[\hat{q}(x_i^k, a)],$$

where $\hat{q}(x, a)$ estimates $q(x, a)$ based on all domains logged bandit data.

**Inverse Propensity Score (IPS).** The IPS estimator, which uses only logged data from the target domain, is defined as follows:

$$\hat{V}_{\mathrm{IPS}}(\pi; \mathcal{D}^T) := \frac{1}{n} \sum_{i=1}^{n} \frac{\pi(a_i|x_i)}{\pi_0^T(a_i|x_i)} r_i = \frac{1}{n} \sum_{i=1}^{n} w^T(x_i, a_i) r_i,$$

where $w^T(x, a) := \pi(a|x)/\pi_0^T(a|x)$ is importance weight of the target domain. The IPS-ALL estimator, which uses logged data from both the target and source domains, is defined as follows:

$$\hat{V}_{\mathrm{IPS-ALL}}(\pi; \mathcal{D}) := \frac{1}{N} \sum_{k=1}^{K} \sum_{i=1}^{n^k} \frac{\pi(a_i^k|x_i^k)}{\pi_0^k(a_i^k|x_i^k)} r_i^k := \frac{1}{N} \sum_{k=1}^{K} \sum_{i=1}^{n^k} w^k(x_i^k, a_i^k) r_i^k,$$

where $w^k(x, a) := \pi(a|x)/\pi_0^k(a|x)$ is importance weight of each domain and $N := \sum_{k=1}^{K} n^k$.

**Doubly Robust (DR).** The DR estimator, which uses only logged data from the target domain, is defined as follows:

$$\hat{V}_{\mathrm{DR}}(\pi; \mathcal{D}^T, \hat{q}^T) := \frac{1}{n} \sum_{i=1}^{n} \{w(x_i, a_i)(r_i - \hat{q}^T(x_i, a_i)) + \mathbb{E}_{\pi(a|x_i)}[\hat{q}^T(x_i, a)]\}.$$

The DR-ALL estimator, which uses logged data from both the target and source domains, is defined as follows:

$$\hat{V}_{\mathrm{DR-ALL}}(\pi; \mathcal{D}, \hat{q}) := \frac{1}{N} \sum_{k=1}^{K} \sum_{i=1}^{n^k} \{w^k(x_i^k, a_i^k)(r_i^k - \hat{q}(x_i^k, a_i^k)) + \mathbb{E}_{\pi(a|x_i^k)}[\hat{q}(x_i^k, a)]\}.$$

### D.2.2 EXPERIMENT RESULTS ON SYNTHETIC DATA

In the following, we report and discuss the MSE, squared bias, and variance of the estimators computed over 150 sets of logged data, each replicated with different seeds. Note that the default parameters in our synthetic validation are $|\mathcal{A}| = 20$, $K = 30$, $n^k = 100$, and $|c^T| = 9$.

First, we compare the estimators under varying ratios of new actions in the target domain, $|\mathcal{U}_0^T|/|\mathcal{A}| \in \{0, 0.2, 0.4, 0.6, 0.8\}$, where $\mathcal{U}_0^T := a \in \mathcal{A} \mid \pi_0^T(a \mid x) = 0, \forall x$ (see Figure 8). The results show that COPE consistently outperforms the other estimators without being affected by the presence of new actions. As in the real-world data experiment, the estimators using only logged data from the target domain, particularly IPS(T) and DR(T), produce a large bias as the ratio increases. Estimators that naively use logged data from all domains, such as IPS(ALL), DR(ALL), and DM(ALL), exhibit consistently high bias due to their inability to handle differences in DGPs.

Second, we compare the estimators under varying percentages of samples with deterministic logging, as shown in Figure 9. In the experiment, we calculate the percentages (20%, 40%, 60%, 80%) of a particular one-dimensional context in the target domain. Then, if the value of the context in the

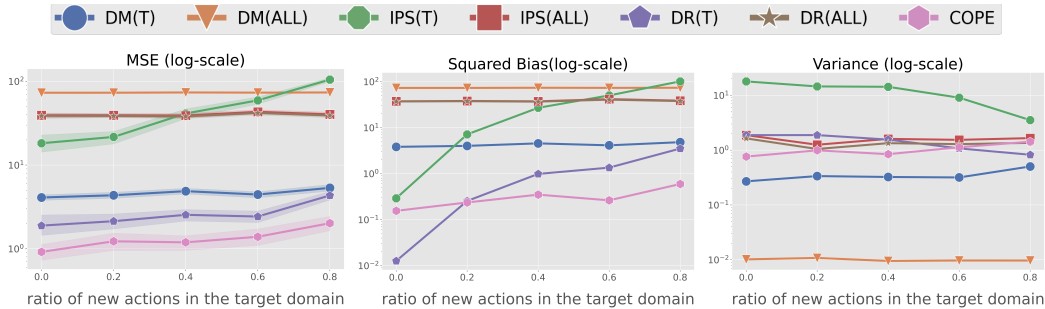

Figure 8: MSE(**left**), Squared Bias(**center**), and Variance(**right**) with varying ratios of new actions in the target domain.

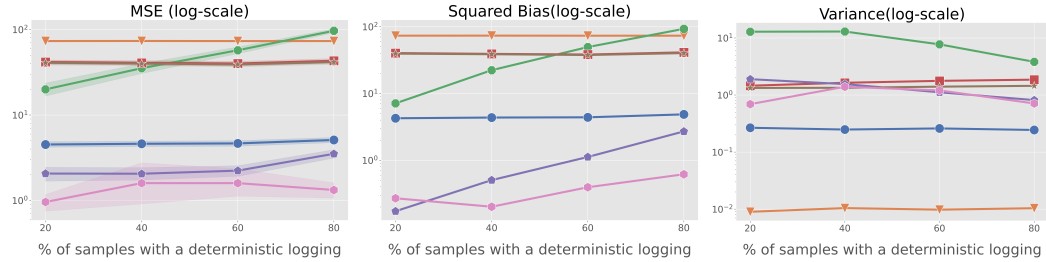

Figure 9: MSE(**left**), Squared Bias(**center**), and Variance(**right**) with varying percentages of samples with a deterministic logging.

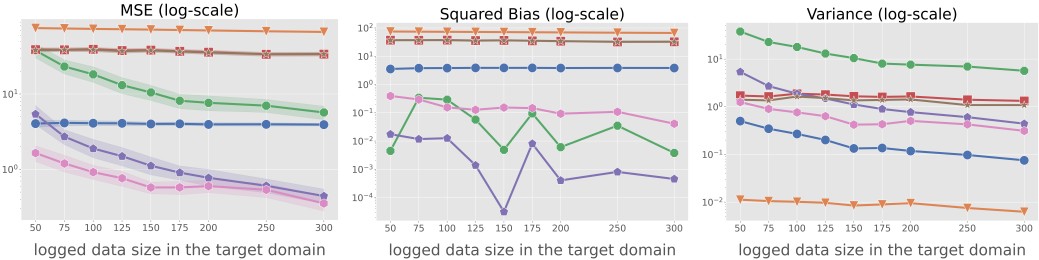

Figure 10: MSE(**left**), Squared Bias(**center**), and Variance(**right**) with varying logged data sizes in the target domain.

target logged data is smaller than the percentile point, we apply a deterministic logging policy; otherwise, we apply a probabilistic logging policy defined by Eq. (16). The results indicate that COPE is robust against an increasing degree of deterministic logging policy and maintains a small bias. In contrast, IPS(T) and DR(T) show increasing bias as the percentile of the target domain context grows, similar to the real-world OPE experiment with varying numbers of users whose logging is deterministic in the target domain.

Finally, we compare the estimators under varying logged data sizes of the target domain $(n^T)$ in Figure 10. We observe that most existing estimators show a decrease in MSE as the logged data size in the target domain increases, but COPE generally achieves the most accurate estimation. Notably, when $n^T = 50$, COPE outperforms the best baseline DM(T) and the second-best baseline DR(T) by a substantial margin $(\frac{\text{MSE}(\hat{V}\text{DM(T)})}{\text{MSE}(\hat{V}\text{COPE})} = 2.46, \frac{\text{MSE}(\hat{V}\text{DR(T)})}{\text{MSE}(\hat{V}\text{COPE})} = 3.29)$.

