# OpenReview forum: "Cross-Domain Off-Policy Evaluation and Learning for Contextual Bandits"
_ICLR.cc/2025/Conference — ICLR 2025 Poster_

### Official Review · Reviewer_A29Z · 2024-10-27

**Soundness:** 3
**Presentation:** 3
**Contribution:** 3
**Rating:** 8
**Confidence:** 4

**Summary:**

This paper tackles the challenging problem of Cross-Domain Off-Policy Evaluation and Learning (OPE/L) for contextual bandits. It focuses on addressing practical issues frequently encountered in real-world applications, such as deterministic logging policies, few-shot data in the target domain, and the presence of new actions. The authors propose COPE, a novel estimator that leverages data from multiple source domains to improve evaluation and learning in a target domain. COPE decomposes expected rewards into domain-cluster and domain-specific effects, aiming to reduce bias and variance. The paper includes an empirical evaluation on a real-world dataset and provides the code for reproducibility.

**Strengths:**

- The paper addresses the important problem in OPE/L for contextual bandits. In particular, the setting where there are deterministic logging policies is often overlooked, despite being a scenario highly relevant to practical deployments. This focus significantly enhances the paper's relevance to real-world applications

- The proposed COPE estimator introduces an interesting approach to leveraging cross-domain data, building upon and extending existing ideas in the field. The connections and distinctions with related estimators, such as OffCEM by Saito et al., are noteworthy

- The empirical evaluation is comprehensive, including ablation studies that provide insights into the contributions of different components of the proposed method. The provided code further strengthens the paper's contribution by ensuring reproducibility

**Weaknesses:**

- While the experimental evaluation on the chosen dataset is comprehensive, I believe the paper would benefit from evaluation on an additional dataset. This would improve the paper, evaluating COPE across different data distributions and characteristics.

- The clarity of the paper could be improved, particularly in explaining the technical details of the COPE estimator, the clustering approach, and the underlying assumptions. More precise definitions and a step-by-step explanation of the methodology would enhance understanding and reproducibility.

- The novelty of the approach is present, however COPE is very similar to OffCEM, as also mentioned in the paper

- The theoretical analysis, which I believe is already interesting, could be strengthened. Specifically, an analysis of the variance of COPE would be valuable. Furthermore, exploring the estimator's bias when the various analyzed conditions are not satisfied would provide a more complete picture of its performance characteristics.

- The theoretical assumptions, such as Conditional Pairwise Correctness (CPC), while providing a foundation for the analysis, might be difficult to verify or guarantee in practice.

Minor:
typo at line 081: Cross-domain Off-Policy EvaluatOIn (COPE)

**Questions:**

- Can the authors provide a theoretical comparison of COPE with the existing methods in the setting of deterministic logging policies?

- Can the authors provide a more detailed explanation of the clustering function, its practical implementation, and its impact on the performance of COPE? What guidelines can be offered for choosing appropriate clustering strategies in different domains?

- What are the practical implications and limitations of the theoretical assumptions in real-world scenarios? How can these assumptions be verified in practice?

- Can the authors provide an analysis of the variance of the COPE estimator? How does the variance compare to existing methods?

- What is the bias of COPE when the conditions for unbiasedness are not met? Is it possible to characterize or bound this bias?

---

> ### Author Response · Authors · 2024-11-15
>
> We appreciate the valuable and thoughtful feedback from the reviewer. We respond to the concrete questions and comments in detail below.
>
> > Can the authors provide a theoretical comparison of COPE with the existing methods in the setting of deterministic logging policies?
>
> This is a good point to clarify. Our current analysis already includes the scenario with deterministic logging as a special case. Specifically, Eq. (3) characterizes the bias of IPS when common support is violated, and when the logging policy is deterministic, nearly all actions are in the set $\mathcal{U}_0$, leading to an extremely large bias for IPS. (Note that it is straightforward to extend Eq. (3) to analyze the bias of DR.)
>
> Even when the logging policy in the target domain is deterministic, it is still possible for Condition 3.1 to be satisfied, allowing the bias of our estimator to be characterized by Theorem 3.1. It is not straightforward to theoretically compare Eq. (3) and Theorem 3.1, so we conducted a comprehensive empirical analysis, demonstrating that our estimator reduces bias compared to IPS(T) and DR(T), particularly for cases with deterministic logging and entirely new actions. This bias reduction is one of the crucial factors to achieve much lower MSE by our estimator.
>
> > Can the authors provide a more detailed explanation of the clustering function, its practical implementation, and its impact on the performance of COPE?
> > What guidelines can be offered for choosing appropriate clustering strategies in different domains?
>
> Thank you for raising this important point for discussion. In our current experiments, we applied KMeans to the empirical average of the rewards in each domain to perform domain clustering, and this heuristic worked effectively and was sufficient to outperform existing methods across a range of experimental setups.
>
> In practice, the number of clusters and the clustering function used can be considered hyperparameters of COPE. We can tune these hyperparameters rigorously by using estimator selection methods for OPE proposed in the following papers.
>
> - Yi Su, Pavithra Srinath, and Akshay Krishnamurthy. Adaptive estimator selection for off-policy evaluation. ICML2020.
> - Takuma Udagawa, Haruka Kiyohara, Yusuke Narita, Yuta Saito, and Kei Tateno. Policy-adaptive estimator selection for off-policy evaluation. AAAI2023.
>
> We will clarify this point further in the revision.
>
> > How can these assumptions be verified in practice?
>
> It appears that the reviewer is concerned about the satisfaction of the CPC condition. It is indeed almost impossible to verify this in practice, which is why we first analyze the bias of COPE when the condition is not satisfied in Theorem 3.1. It is also important to note that in challenging cases, such as deterministic logging and the presence of new actions, every existing estimator, including IPS and DR, produces much bias, but our estimator leverages source domain data and is much more robust to the bias arising due to those challenges. We conducted comprehensive experiments to compare the MSE, bias, and variance of the estimators across a range of challenging cases, and **we demonstrated that our estimator performed best in most scenarios even with an estimated reward function ($\hat{q}$) that likely violates the condition**.
>
> > Can the authors provide an analysis of the variance of the COPE estimator? How does the variance compare to existing methods? What is the bias of COPE when the conditions for unbiasedness are not met?
>
> Thank you for suggesting interesting analyses. First, it is important to note that Theorem 3.1 already provides the bias of COPE when the CPC condition is not satisfied. It is also straightforward to analyze the bias of COPE under violations of Condition 3.1 in a manner similar to Eq. (3). Additionally, it is possible to analyze the variance of COPE. Intuitively, COPE has lower variance than IPS and DR due to the additional use of source domain data. Furthermore, COPE has lower variance than IPS-ALL and DR-ALL due to the use of multiple importance weighting techniques, as already shown in (Agarwal et al., 2017; Kallus et al., 2021). The variance reduction effect of COPE has also been empirically demonstrated in our experiments as well.

---

> > ### Comment · Reviewer_A29Z · 2024-11-22
> > **Thank you**
> >
> > Thanks for the response.
> >
> > I agree with your points.
> > I just want to see the revised paper with the new experiments you were asked to perform by Reviewer KgjZ before deciding whether to update my score though.

---

> > > ### Author Response · Authors · 2024-11-26
> > >
> > > We appreciate the timely and thoughtful discussion provided by the reviewer once again.
> > >
> > > We would now like to share additional experimental results obtained from the ablation study we performed based on Reviewer KgjZ’s suggestion. Specifically, we exclude $\bar{q}(u)$ from the definition of the user distribution $p^k(u)$, following Reviewer KgjZ’s observation that this might favor our methods. Instead, we define a linear function of the user feature $f^k(u) = \beta^k x_u$, where the coefficient vector $\beta^k$ is sampled from a normal distribution separately for each domain $k$. We then define the user distribution $p^k(u)$ for each domain as $ p^{k}(u):=\frac{\exp (f^k(u))}{\sum_{u^{\prime} \in \mathcal{U}} \exp (f^k(u^{\prime}))} $. This procedure follows the multi-domain recommendation literature, which uses user features to define the domains in their experiments [Yang et al. 2024]. More importantly, this approach avoids any dependence of the distribution on $\bar{q}(u)$. We conducted both OPE and OPL experiments based on this new user distribution, and the results are summarized in the following tables.
> > >
> > > ---
> > >
> > > **MSE in OPE for varying ratios of new actions in the target domain** (values are smaller the better)
> > > | ratios of new actions | 0.0    | 0.2    | 0.4    | 0.6    | 0.8    |
> > > |-----------|--------|--------|--------|--------|--------|
> > > | **COPE(Ours)**       | **1.1641** | **1.1647** | **0.8066** | **1.0663** | **1.0801** |
> > > | DR(Target Domain)        | 4.3283 | 4.2225 | 2.3985 | 2.9498 | 3.7964 |
> > > | DR(ALL Domain)    | 2.7582 | 2.2401 | 3.5354 | 3.3172 | 1.4530 |
> > > | IPS(Target Domain)       | 5.6442 | 4.9797 | 3.6800 | 4.8295 | 6.5302 |
> > > | IPS(ALL Domain)   | 2.1763 | 2.2538 | 2.2006 | 2.1881 | 2.9360 |
> > >
> > > **MSE in OPE for varying numbers of users with deterministic logging in the target domain** (values are smaller the better)
> > >
> > > | numbers of users with deterministic logging | 200    | 400    | 600    | 800    | 1000   |
> > > |-----------|--------|--------|--------|--------|--------|
> > > | **COPE(Ours)**      | **1.3238** | **1.3120** | **1.2926** | **1.2125** | **1.2172** |
> > > | DR(Target Domain)        | 2.3200 | 2.8398 | 3.2755 | 3.7992 | 4.4499 |
> > > | DR(ALL Domain)    | 2.5240 | 2.7915 | 3.0358 | 2.9843 | 3.3314 |
> > > | IPS(Target Domain)       | 3.2026 | 3.9641 | 4.6112 | 5.6947 | 7.0049 |
> > > | IPS(ALL Domain)   | 2.1561 | 2.1630 | 2.1906 | 2.2188 | 2.2076 |
> > >
> > >
> > > **Test (Relative) Policy Value in OPL for varying ratios of new actions in the target domain** (values are larger the better)
> > > | ratios of new actions    | 0.0    | 0.2    | 0.4    | 0.6    | 0.8    |
> > > |--------------|--------|--------|--------|--------|--------|
> > > | **COPE-PG(Ours)**      | **1.4463** | **1.4060** | **1.4389** | **1.3884** | 1.4090 |
> > > | DR-PG(Target Domain)        | 1.3518 | 1.3220 | 1.3364 | 1.2663 | 1.3339 |
> > > | DR-PG(ALL Domain)    | 1.4308 | 1.3947 | 1.4064 | 1.3420 | **1.4403** |
> > > | IPS-PG(Target Domain)        | 1.3119 | 1.2755 | 1.2899 | 1.2389 | 1.3350 |
> > > | IPS-PG(ALL Domain)   | 1.3588 | 1.3198 | 1.3267 | 1.3134 | 1.3246 |
> > >
> > >
> > > **Test (Relative) Policy Value in OPL for varying numbers of users with deterministic logging in the target domain** (values are larger the better)
> > > | numbers of users with deterministic logging    | 200    | 400    | 600    | 800    | 1000   |
> > > |--------------|--------|--------|--------|--------|--------|
> > > | **COPE-PG(Ours)**      | **1.4408** | **1.4531** | **1.4769** | **1.4669** | 1.4139 |
> > > | DR-PG(ALL Domain)   | **1.4477** | 1.4180 | 1.3954 | 1.4059 | **1.5115** |
> > > | DR-PG(Target Domain)     | 1.3101 | 1.3014 | 1.3091 | 1.3596 | 1.3013 |
> > > | IPS-PG(ALL Domain)  | 1.3381 | 1.3003 | 1.3149 | 1.3625 | 1.4651 |
> > > | IPS-PG(Target Domain)    | 1.2795 | 1.2770 | 1.2905 | 1.2765 | 1.2467 |
> > >
> > > ---
> > >
> > > **The results indicate that our method mostly outperforms the baseline methods that use only the target domain data or all the data across target and source domains in both OPE and OPL experiments** (bold fonts indicate the best method, if the best and second best are not significantly different, we use bold fonts for both).

---

> > > > ### Author Response · Authors · 2024-11-26
> > > >
> > > > Furthermore, we can extend the definition of the user distribution as:
> > > >
> > > > $$ p^{k}(u):=\frac{\exp (\lambda \bar{q}(u) + (1-\lambda) f^k(u))}{\sum_{u^{\prime} \in \mathcal{U}} \exp (\lambda \bar{q}(u’) + (1-\lambda) f^k(u')))} $$
> > > >
> > > > where $\lambda$ controls the degree to which the user distribution depends on $\bar{q}(u)$. When $\lambda = 1$, the setup reduces to our original experiment. The following reports the OPL experiment with varying $\lambda$ within $\\{ 0.0, 0.2, 0.4, 0.6, 0.8, 1.0 \\}$.
> > > >
> > > >
> > > > **Test (Relative) Policy Value in OPL for varying $\lambda$ values** (values are larger the better)
> > > > | lambda values   | 0.0   | 0.2   | 0.4   | 0.6   | 0.8   | 1.0   |
> > > > |------------|-----------|-----------|-----------|-----------|-----------|-----------|
> > > > | **COPE(Ours)**  | **1.4713**    | **1.4646**    | **1.4739**    | **1.4502**    | **1.4780** | **1.4404** |
> > > > | DR-PG(Target Domain) | 1.3812    | 1.3558    | 1.3475    | 1.3592    | 1.3042    | 1.3657    |
> > > > | DR-PG (ALL Domain) | 1.4639    | 1.4334    | 1.3823    | 1.4374    | 1.3956    | 1.2799    |
> > > > | IPS-PG(Target Domain)| 1.3553    | 1.3010    | 1.3133    | 1.2869    | 1.2770    | 1.2755    |
> > > > | IPS-PG (ALL Domain) | 1.3629    | 1.3580    | 1.3572    | 1.3480    | 1.4390    | 1.3889    |
> > > >
> > > > We should mention here that the difference between the proposed method and the baselines decreases when $\lambda$ is small, which validates Reviewer KgjZ’s comment. **These additional observations, however, demonstrate the robustness of COPE to varying user distributions, even with a simple clustering procedure.** Nevertheless, we believe a more principled and effective approach to domain clustering for our method likely exists. As mentioned in Section 5, we consider the development of a clustering algorithm tailored to Cross-Domain OPE and OPL to be a valuable direction for future research.
> > > >
> > > > As a side note, we have updated our draft to include clarifications on points discussed with the reviewers, such as data-driven tuning of cluster numbers, bias comparisons against IPS/DR-ALL, density ratio estimation, and the convergence properties of COPE-PG under the CPC condition. We have also clarified that “deterministic logging” and “new actions” can be interpreted as harder versions of the common support violation to avoid confusion.   These updates are highlighted in red in the text. **We would greatly appreciate it if the reviewer could confirm whether the additional results address their main concern or if there are any remaining issues that require further discussion.**
> > > >
> > > > ---
> > > >
> > > > [Yang et al. 2024] Zhiming Yang, Haining Gao, Dehong Gao, Luwei Yang, Libin Yang, Xiaoyan Cai, Wei Ning, Guannan Zhang. MLoRA: Multi-Domain Low-Rank Adaptive Network for CTR Prediction. RecSys2024.

---

> > > > > ### Comment · Reviewer_A29Z · 2024-11-26
> > > > > **Thanks**
> > > > >
> > > > > Thank you for your response. After seeing these experiments and the discussion, I've updated my score.

---

> > > > > > ### Author Response · Authors · 2024-11-27
> > > > > >
> > > > > > We appreciate the reviewer for carefully evaluating our additional experiments and for all the effort they have put into rigorously understanding our contributions. We will summarize the additional results in the appendix and mention it in the main text. We will then update the draft by the deadline.
> > > > > >
> > > > > > If the reviewer has any remaining questions or concerns about our work, we would be more than happy to discuss them further.

---

### Official Review · Reviewer_8D6L · 2024-10-29

**Soundness:** 3
**Presentation:** 2
**Contribution:** 2
**Rating:** 6
**Confidence:** 2

**Summary:**

This paper focuses on addressing the problem of OPE/L when the common support assumption is not met. It introduces a new estimator based on the Common Cluster Support assumption and multi-source data to tackle this issue. The proposed approach is validated through semi-simulated experiments.

**Strengths:**

- The challenges of OPE/L with multi-source data and violations of the common support assumption are significant and relevant in practical contexts.

- The proposed estimator aligns well with the motivations of this paper and provides theoretical support for its unbiasedness.

**Weaknesses:**

- This paper does not compare with significant literature focused on the limited overlap problem. For instance, the assumptions in Condition 3.1 and the subsequent proof of unbiasedness are fundamentally similar to those in [1, 2], which also assume that while the original x may not satisfy the overlap assumption, there exists a score or representation that does. The unique contribution of this paper lies in its application to the cross-domain OPE/L setting.

- However, the absence of a real cross-domain experimental validation, relying instead on semi-simulated experiments, limits the contribution of this work.

If the authors clearly outline the unique challenges of limited overlap in the context of multi-source data compared to conventional scenarios, and explain the specific efforts made to address these challenges, I would consider increasing my score.
Additionally, if the authors provide evaluation results of their method in real-world settings, I would also reconsider my score.

---
[1] Hansen B B. The prognostic analogue of the propensity score[J]. Biometrika, 2008, 95(2): 481-488.

[2] Wu P, Fukumizu K. $\beta $-Intact-VAE: Identifying and Estimating Causal Effects under Limited Overlap[J]. arXiv preprint arXiv:2110.05225, 2021.

**Questions:**

see above

---

> ### Author Response · Authors · 2024-11-15
>
> We appreciate the valuable and thoughtful feedback from the reviewer. We respond to the concrete questions and comments in detail below.
>
> > This paper does not compare with significant literature focused on the limited overlap problem. For instance, the assumptions in Condition 3.1 and the subsequent proof of unbiasedness are fundamentally similar to those in [1, 2], which also assume that while the original x may not satisfy the overlap assumption, there exists a score or representation that does.
>
> Thank you for providing these valuable references; we will cite them in the revised version of the paper. **It is important to note, however, that the problem setup and methods proposed in the papers suggested by the reviewer are substantially different from ours and are not directly comparable**.
>
> First, the papers suggested by the reviewer focus on causal effect estimation, while we study off-policy evaluation and learning of contextual bandit policies. In particular, the policy learning problem is not considered at all in those previous works.
>
> Second, the papers suggested by the reviewer address limited overlap in $x$, whereas we deal with the limited overlap regarding the action $a$. We even tackle the problem of completely new actions, which is a distinct and unique aspect of our work.
>
> Third, we address the issue of limited overlap in $a$ by leveraging structure across different domains, while the papers suggested by the reviewer leverage structure within a single domain. Indeed, we could not find the term “cross-domain” in the text of these papers, which indicates that our approach and the prognostic score approach in these works could potentially be combined. This suggests that our work and these previous works are orthogonal to each other.
>
> As discussed here, our motivation, formulation, and approach are substantially different from those of previous work. We will clarify this distinction in the revision.
>
>
> > the absence of a real cross-domain experimental validation, relying instead on semi-simulated experiments, limits the contribution of this work.
>
> Thank you for raising this important point. We agree that it would be ideal to conduct empirical evaluation on a real-world cross-domain dataset; however, there is currently no public dataset available that enables such an experiment in the off-policy bandit setup. **It is indeed standard to perform fully synthetic or semi-synthetic experiments in most of the relevant papers we discussed in Section 1 on page 2, and this limitation applies not only to our work but also to most other related studies**. We will clarify this point in the revision.

---

> > ### Comment · Reviewer_8D6L · 2024-11-21
> >
> > Thank you for your detailed response. Your comments appear to corroborate my review:
> > - The authors' suggestion that "the papers addressing limited overlap in $x$ differ from our approach dealing with limited overlap regarding the action $a$" partially supports the first weakness in my review that the unique contribution of this paper lies in the introduction of the limited overlap to the cross-domain OPE/L setting.
> > - The acknowledgment that "currently no public dataset exists to enable experiments in the off-policy bandit setup" further substantiates the second weakness that potentially limits the work's contribution.
> >
> > Consequently, I will maintain my original assessment.

---

> > > ### Author Response · Authors · 2024-11-21
> > >
> > > We would like to thank the reviewer for reading our responses and providing the additional comments. However, we would like to kindly argue that some parts of them are not very convincing.
> > >
> > > > The authors' suggestion that "the papers addressing limited overlap in differ from our approach dealing with limited overlap regarding the action $a$ " partially supports the first weakness in my review that the unique contribution of this paper lies in the introduction of the limited overlap to the cross-domain OPE/L setting.
> > >
> > > We respectfully but strongly disagree with this comment. **It appears that the reviewer has overlooked the critical contributions of our work, which are addressing deterministic logging policies and completely new actions in OPE/L by newly formulating the cross-domain setup.** **These problems cannot be addressed by the papers cited by the reviewer, as the covariate shift in $x$ is orthogonal to the issues regarding the logging policy $\pi_0(a|x)$, a distinction correctly understood by the other reviewers**.
> > >
> > > It is surprising that the reviewer believes our contributions lack novelty simply because we are addressing limited overlap in $a$ not only $x$. If that is the case, then by the same logic, every paper proposing variants of importance weighting or inverse propensity scoring (such as [1-12] below) would lack novelty due to the existence of the covariate shift problem. We disagree with this perspective, as $x$ originates from the environment, whereas $a$ is sampled from a conditional policy distribution $\pi(a|x)$. **This results in the critical differences between our work and the papers shared by the reviewer, i.e., our motivations are tackling deterministic $\pi_0$ and completely new $a$, while the papers shared by the papers cannot and does not aim to address deterministic $p(x)$ and completely new $x$.** It is also important to remember that the papers mentioned by the reviewer do NOT work on the policy learning side at all, which are one of our main motivations as well.
> > >
> > > **If the reviewer still believes that the problems and critical issues we are addressing (such as deterministic $p(x)$ and completely new $x$ as we did regarding $a$) can be solved by the papers they have suggested, we would kindly request a specific demonstration of how this could be achieved**. We would be eager to learn such an extremely innovative extension if it is indeed possible. **However, if this is not the case, it is difficult to agree with the argument that our contributions lack novelty simply due to the existence of papers that focus on substantially different motivations and problems**.
> > >
> > > > "currently no public dataset exists to enable experiments in the off-policy bandit setup" further substantiates the second weakness that potentially limits the work's contribution.
> > >
> > > **We would like to emphasize that the KuaiRec dataset is the only publicly available dataset that provides the full reward matrix, which is a crucial requirement for conducting experiments related to OPE and OPL. In fact, dozens of papers accepted at top-tier relevant conferences rely on similarly synthetic or even more synthetic experiments [1–12], just to name a few**. We agree that it would be ideal, as a community, to construct more real-world public datasets that enable rigorous experiments on OPE and OPL, but this is a totally orthogonal motivation to our work.
> > >
> > > If the reviewer still believes that our experiments do not adhere to best practices in the relevant community, **we would appreciate it if the reviewer could specify the datasets and experimental designs they consider appropriate for our work. If the reviewer is unable to provide such specific resources and designs, we would have to conclude that this comment lacks specificity and does not accurately reflect the current state and practices of the OPE/L research community.** However, we believe this is not the case and that the reviewer likely has specific datasets and designs in mind as a background of the comment. We would be happy to discuss these further.

---

> > > > ### Author Response · Authors · 2024-11-21
> > > >
> > > > Here are the references we used in the response:
> > > >
> > > > [1] Yuta Saito, Qingyang Ren, Thorsten Joachims. Off-Policy Evaluation for Large Action Spaces via Conjunct Effect Modeling. ICML2023.
> > > >
> > > > [2] Yi Su*, Lequn Wang*, Michele Santacatterina, Thorsten Joachims. CAB: Continuous Adaptive Blending for Policy Evaluation and Learning. ICML2019.
> > > >
> > > > [3] Muhammad Faaiz Taufiq, Arnaud Doucet, Rob Cornish, Jean-Francois Ton. Marginal Density Ratio for Off-Policy Evaluation in Contextual Bandits. NeurIPS2023.
> > > >
> > > > [4]  Haruka Kiyohara, Masahiro Nomura, Yuta Saito. Off-Policy Evaluation of Slate Bandit Policies via Optimizing Abstraction. WWW2024.
> > > >
> > > > [5] Noveen Sachdeva*, Yi Su*, Thorsten Joachims. Off-policy Bandits with Deficient Support. KDD2020.
> > > >
> > > > [6] Tatsuhiro Shimizu, Koichi Tanaka, Ren Kishimoto, Haruka Kiyohara, Masahiro Nomura, Yuta Saito. Effective Off-Policy Evaluation and Learning in Contextual Combinatorial Bandits. RecSys2024.
> > > >
> > > > [7] Yi Su, Pavithra Srinath, Akshay Krishnamurthy. Adaptive Estimator Selection for Off-Policy Evaluation. ICML2020.
> > > >
> > > > [8] Allen Nie, Yash Chandak, Christina J. Yuan, Anirudhan Badrinath, Yannis Flet-Berliac, Emma Brunskil. OPERA: Automatic Offline Policy Evaluation with Re-weighted Aggregates of Multiple Estimators. NeurIPS2024.
> > > >
> > > > [9] Takuma Udagawa, Haruka Kiyohara, Yusuke Narita, Yuta Saito, Kei Tateno. Policy-Adaptive Estimator Selection for Off-Policy Evaluation. AAAI2023.
> > > >
> > > > [10] Nicolò Felicioni, Michael Benigni, Maurizio Ferrari Dacrema. Automated Off-Policy Estimator Selection via Supervised Learning. https://arxiv.org/abs/2406.18022
> > > >
> > > > [11] Masahiro Kato, Masatoshi Uehara, Shota Yasui. Off-Policy Evaluation and Learning for External Validity under a Covariate Shift. NeurIPS2020.
> > > >
> > > > [12] Zhengyuan Zhou, Susan Athey, Stefan Wager. Offline Multi-Action Policy Learning: Generalization and Optimization. https://arxiv.org/abs/1810.04778.

---

> > > > ### Comment · Reviewer_8D6L · 2024-11-22
> > > >
> > > > I appreciate your comprehensive response regarding the two identified weaknesses. Let me address each point specifically.
> > > >
> > > > - Regarding the Limited Overlap Definition: You appear to have misinterpreted the concept of limited overlap. The setting is formally defined where $p(t|x)>0$ does not hold universally, where $t$ denotes treatment. The downstream objective of causal effect estimation explicitly addresses the treatment assignment policy and new treatments $t$. This directly parallels the OPL setting with $\pi_0(a|x)$. Therefore, this work essentially translates the limited overlap problem framework to the OPL context. Your reference to IPS demonstrates an understanding that causal effect estimation and OPL methodologies share common ground, both focusing on learning the policy $\pi_0(a|x)$.
> > > > - Regarding Previous Literature [1-12]: The cited works primarily address challenges in adapting IPS to OPL:  Papers [1-4, 6] tackle high variance issues; Paper [5] addresses limited overlap; Papers [7-10] focus on estimator selection; Papers [11-12] provide novel theoretical guarantees.
> > > > These works develop specific algorithms for traditional OPL setting challenges. Your paper differs by proposing a novel cross-domain OPE/L setting to address limited overlap, rather than solely offering methodological solutions. Consequently, the validity of your approach depends not only on algorithmic soundness but also on the applicability of the cross-domain OPE/L setting.
> > > > - Regarding Real-world Data Validation:
> > > > Consider examining multi-domain recommendation literature, which utilizes real-world datasets (e.g., MovieLens-1M, KuaiRand-Pure). Evaluation metrics can be diversified beyond semi-synthetic ground truth to include proxy indicators (e.g., total gains, AUC) that demonstrate real-world utility.
> > > > - Summary: A methodology addressing deficiencies in previous limited overlap approaches within traditional OPE/L settings would be compelling. However, proposing a cross-domain OPE/L setting validated only on semi-synthetic data potentially limits the work's contribution.
> > > > - Suggestion: I understand the author's eagerness to defend their work. However, I encourage the author to carefully verify whether their response directly addresses the weaknesses identified by the reviewers to ensure the effectiveness of the discussion.

---

> > > > > ### Author Response · Authors · 2024-11-27
> > > > >
> > > > > We would like to thank the reviewer for their detailed response.
> > > > >
> > > > > To begin with, it seems that the reviewer assumes we are discussing to defend our work, but this is not the case. We greatly appreciate and value constructive reviews and convincing/concrete suggestions, even if the paper is rejected.
> > > > >
> > > > > For example, Reviewer KgjZ (whose initial score was 3) has provided reviews and suggestions that are highly effective. Reviewer KgjZ has proposed a very concrete additional experiment based on their interesting observation. The suggestion was so convincing and detailed that it immediately motivated us to perform the recommended experiment, which ultimately improved our contributions. We truly appreciate such a thoughtful and detailed suggestion, regardless of their score.
> > > > >
> > > > > Since the reviewer has kindly clarified some of their previous comments, we believe we can now begin a more in-depth discussion below.
> > > > >
> > > > > > Regarding the Limited Overlap Definition: You appear to have misinterpreted the concept of limited overlap…
> > > > >
> > > > > We appreciate the reviewer’s concrete description of their current understanding. It allows us to identify where we should clarify further.
> > > > >
> > > > > First of all, we understand what limited overlap generally means. In the context of typical OPE/L, it refers to the violation of the common support condition (as already framed as Condition 2.1), which aligns with the reviewer’s description in their response. This is called “deficient support” in the context of OPE/L [Sachdeva et al. 2020]. This corresponds to the violation of Definition 1 in [2], a paper suggested by the reviewer.
> > > > >
> > > > > Note that limited overlap or deficient support still allow the logging policy $\pi_0$ to remain stochastic for some $x$. However, especially in industry problems such as ad placement, recommender, and search systems, logging policy is often completely deterministic, mostly due to the difficulty of performing stochastic exploration in large-scale real systems. It is also true that we need to perform OPE and OPL in the presence of new actions. For example, in the news recommendation problem, a ranking system(policy) needs to deal with completely new articles coming into the system day by day, and those actions are not observed in the logged data at all for any feature simply because they are new.
> > > > >
> > > > > Our main motivations are to address the challenges of such deterministic logging and new actions by newly leveraging cross-domain data. These scenarios can be interpreted as harder instances of the general limited overlap or deficient support problem.
> > > > >
> > > > > - **Deterministic logging policy** refers to the scenario where $\pi_0(a|x) = 1$ for a certain $a \in \mathcal{A}$ and $\pi_0(a'|x) = 0, \forall a' \in \mathcal{A} \backslash \\{a \\}$. This condition is sufficient but not necessary for the standard limited overlap because the policy is completely deterministic. Translating this into a binary treatment setup, this would mean $\pi_0(t|x) = 1$ for either $t=0$ or $t=1$ for all $x$. This is different from the problem setups and experiments considered in the papers suggested by the reviewer (e.g., [2] uses $\pi_0(t|x) > 0$ for all $x$ and $t$ in their synthetic experiment though the probabilities can sometimes be very small. The logging policy(RCT) of the IHDP dataset used in [2] is also mostly stochastic rather than deterministic).
> > > > >
> > > > >
> > > > > - **Completely new actions** refer to actions $a$ that have zero probability of being selected under the logging policy across all features, i.e., $\pi_0(a|x) = 0, \forall x \in \mathcal{X}$. These actions are completely new, as they cannot be observed in the logged data. This is also a sufficient but not necessary condition for the standard limited overlap problem. In the binary treatment setup, “completely new action” corresponds to the scenario where $\pi_0(t=0|x) = 1$ for all $x$, and we aim to evaluate and learn new policies involving $t=1$, which is completely unobserved in the logged data for any feature. This is definitely not the focus of [1] and [2]. In contrast, in a more general contextual bandit setup, we develop methods to evaluate and optimize new policies even with new actions in the target domain by leveraging source domain data. (note that we have already empirically compared and outperformed naive approaches of combining target and source domain data in the presence of new actions.)

---

> ### Author Response · Authors · 2024-11-27
>
> The reviewer thankfully shared a few papers in the initial review, but we are confident that their methods do not aim to address the specific challenges that our methods tackle (deterministic logging and new actions). It should be noted that the other reviewers explicitly reference “deterministic logging” and “new actions” in their reviews or discussion, while the reviewer states only:
>
> > This paper focuses on addressing the problem of OPE/L when the common support assumption is not met.
>
> Thus, it is unclear if the reviewer is aware of the difference between the mere violation of common support and its harder scenarios of deterministic logging and new actions. It is crucial to ensure the reviewer fully understands our key motivations and contributions, as the other reviewers thankfully do.
>
> We have revised our text to improve clarity regarding deterministic logging and new actions and to emphasize that these represent harder versions of the typical limited overlap or deficient support problem (the updates are highlighted in red in the text).
>
> > Regarding Real-world Data Validation: Consider examining multi-domain recommendation literature, which utilizes real-world datasets (e.g., MovieLens-1M, KuaiRand-Pure). Evaluation metrics can be diversified beyond semi-synthetic ground truth to include proxy indicators (e.g., total gains, AUC) that demonstrate real-world utility.
>
> Thank you for suggesting the datasets and metrics. We have considered using them, however, it is unclear what specific value these datasets and metrics add to our research.
>
> First, it is important to note that the MovieLens dataset does not provide a full-reward matrix, which is crucial for performing OPE/L without synthesizing the reward function, as described in [Gao et al. 2022]. Additionally, we looked at one of the latest papers on cross-domain recommendations [Yang et al. 2024]. The paper uses MovieLens and defines different domains based on user features (e.g., gender), which is essentially the same approach we use to produce domains in our experiments. Given that the MovieLens dataset lacks a full-reward matrix and thus we need to synthesize it, we are unclear how an additional experiment using MovieLens produces value compared to our current experiments.
>
> Second, the reviewer suggested using the KuaiRand dataset. We are similarly confused with this suggestion, because KuaiRand is a dataset targeted for the task of sequential recommendation collected on the Kuaishou app. We have already used the KuaiRec dataset (https://kuairec.com/) that was collected on exactly the same app, and is more suitable for our setup of OPE/L in contextual bandits. We agree that KuaiRand is more suitable if we are focusing on a sequential problem such as reinforcement learning, but this is indeed not the case for our work. In addition, KuaiRand does not provide multiple domains, meaning we would need to generate domains ourselves, as we have already done in our experiments or as done in [Yang et al. 2024]. Therefore, we are not yet sure why using the KuaiRand dataset would add value to our research, given that we have already used the KuaiRec dataset from the same platform.
>
> If the reviewer has more specific experimental designs for using these datasets to produce more advantageous and insightful experiments for the specific problem of cross-domain OPE/OPL compared to what we have provided, we would need to understand those designs in detail to have a more constructive discussion.
>
> Additionally, the reviewer suggested a few proxy indicators, but it is unclear what is meant by “total gains.” We would appreciate a rigorous definition to discuss if it would be meaningful to add it as a reward in our experiments. We are also confused by the suggestion of using AUC, as it is a ranking or classification metric. Given that the reviewer is proposing a metric that is irrelevant to our off-policy learning setup with a single action, there may be a misunderstanding that we are addressing a ranking/classification problem and aiming to maximize an AUC-type metric, which does not align with our formulation. Regarding this context, we think it is important to note that the issues we are tackling are agnostic to how we define the reward. Therefore, we would need to understand why the reviewer thinks it is important to test various reward definitions, even though it is a more general problem and is not directly relevant to our focuses such as deterministic logging, new actions, and the cross-domain formulation.

---

> ### Author Response · Authors · 2024-11-27
>
> Here are the references we used in the response:
>
> ---
>
> [1] Hansen B B. The prognostic analogue of the propensity score[J]. Biometrika, 2008, 95(2): 481-488.
>
> [2] Wu P, Fukumizu K. beta-Intact-VAE: Identifying and Estimating Causal Effects under Limited Overlap[J]. arXiv preprint arXiv:2110.05225, 2021.
>
> [Sachdeva et al. 2020]: Noveen Sachdeva, Yi Su, Thorsten Joachims. Off-policy Bandits with Deficient Support. KDD2020.
>
> [Gao et al. 2022]: Chongming Gao, Shijun Li, Wenqiang Lei, Jiawei Chen, Biao Li, Peng Jiang, Xiangnan He, Jiaxin Mao, Tat-Seng Chua. KuaiRec: A Fully-observed Dataset and Insights for Evaluating Recommender Systems. CIKM2022.
>
> [Yang et al. 2024] Zhiming Yang, Haining Gao, Dehong Gao, Luwei Yang, Libin Yang, Xiaoyan Cai, Wei Ning, Guannan Zhang. MLoRA: Multi-Domain Low-Rank Adaptive Network for CTR Prediction. RecSys2024.

---

> > ### Comment · Reviewer_8D6L · 2024-11-28
> >
> > I appreciate the effort you put into revising the paper, especially within the limited time available.
> >
> > - The revised version clearly articulates the specific challenges addressed by the work in tackling the limited overlap/deficient support problem. This is indeed a uniquely challenging aspect of the limited overlap issue, and I believe the paper has effectively resolved the first identified weakness I concerned. Based on these improvements, I will increase my evaluation score.
> >
> > - Regarding experiments, my suggestion has always been to conduct evaluations entirely based on real-world datasets rather than relying on semi-synthetic experiments. This would allow for assessing the proposed policies against actual business outcomes. However, it seems there was some misunderstanding, as the authors interpreted my suggestion as a request for experiments using datasets like MovieLens-1M or KuaiRand-Pure. This was not my intention, and I apologize for any ambiguity that led to this misinterpretation. Experiments using the original KuaiRec dataset would also be suitable for this purpose.
> > As for evaluation metrics, the "total gains" metric could be defined by the authors to reflect outcomes such as overall click-through rates or customer retention rates. While I understand your concern that focusing on AUC might give readers the impression of addressing a specific business context, you could broaden the evaluation by incorporating additional business-oriented metrics. Furthermore, you are not limited to using business data alone. If you can identify multi-source datasets from domains such as healthcare, education, or other fields for direct experimentation, it would also address my concerns effectively.  Including these assessments in an appendix or discussion section could highlight the cross-domain applicability of OPE/L and underscore the practical value of your method.

---

> > > ### Author Response · Authors · 2024-11-28
> > >
> > > We appreciate the reviewer’s effort in carefully evaluating our work. We are confident that the reviewer’s suggestions have improved our contributions and presentations, and it has been a valuable learning experience to engage in discussions with the reviewer.
> > >
> > > We would also like to thank the reviewer for clarifying their previous comments. First, we appreciate the reviewer acknowledging our contributions and the distinctions between our work and existing studies, particularly regarding issues such as deterministic logging and new actions. Second, we now understand that the reviewer did not necessarily argue for the use of MovieLens-1M or KuaiRand. We also appreciate the reviewer recognizing the plausibility of the KuaiRec dataset we used. In addition, we would like to note that the reward metric used in our current experiments is the watch ratio and we measure the policy performance by the expected watch ratio under its deployment, which we believe is business-oriented and may already align with the reviewer’s concept of “total gains.”
> > >
> > > We fully agree that it would be ideal to use completely real-world datasets for experiments. However, it is also true that a full-reward matrix is required for such experiments, and currently, only the KuaiRec dataset provides this capability. We will more explicitly mention this as a potential limitation of our work. Thank you once again for the valuable comments.

---

### Official Review · Reviewer_KgjZ · 2024-11-02

**Soundness:** 3
**Presentation:** 3
**Contribution:** 3
**Rating:** 6
**Confidence:** 4

**Summary:**

This work studies off-policy estimation and learning (OPE/OPL) for the cross-domain scenario, where logged bandit feedback for different domains is leveraged to improve the offline estimation and learning for policies on a target domain. The proposed estimator COPE relies on a reward decomposition inspired by previous work. Its bias and variance were studied for OPE and a straightforward policy gradient approach was derived for OPL. Experiments show that the estimator performs well on cross-domain scenarios with support deficiency in both OPE and OPL

**Strengths:**

- The paper is well written and pleasant to read.
- The experimental section shows favorable results for the proposed solution.

**Weaknesses:**

- If the cross-domain application in OPE/OPL is novel, the motivation and contribution of the work is oversold:
   + Hospitals for example rarely share data of patients with centralised entities, and learning on this type of sensitive data is primarily done within the framework of _federated learning_ [3]. The described framework where the learner has access to the data of all hospitals in a centralised manner is far from reality.
   + The MIPS [1] and OffCEM [2] estimator for example are based on the same reward decomposition, and treat the large action space scenario, where support deficiency happens all the time, as the logging policy cannot cover all the action space. For example, this sentence in the work: "However, our work differs from these studies in motivation, as we formulate the cross-domain OPE/OPL problem to solve non-trivial challenges such as new actions and deterministic logging" lacks support, as MIPS and OffCEM already solve these non-trivial challenges. See for example Section 3 of [1] where one of the main motivations is going beyond the common support assumption.

- The derivation technique of the estimator is not novel. The COPE estimator is based on the same technique of [1, 2], with little adjustment  to the new application. Its bias/variance tradeoff is also studied in the same way as [2], which makes the theoretical contribution unoriginal.


- The work claims to study OPL, which was not done in [1, 2]. Their proposed OPL method is based on the derivation of a policy gradient, which is basically computing the gradient of the proposed estimator and using it in first order optimisation methods. This contribution is also straightforward and lacks studying in the paper.

- The paper omits a large spectrum of OPE/OPL contributions based on the pessimism principle. This principle was heavily studied for OPL specifically and was proven to be optimal, contrary to optimising the estimator directly. The following lines of work should be incorporated to the related work and discussed, the following papers can be a good start:

   + Bayesian Counterfactual Risk Minimization. Ben London, Ted Sandler Proceedings of the 36th International Conference on Machine Learning, PMLR 97:4125-4133, 2019.
   + Confident Off-Policy Evaluation and Selection through Self-Normalized Importance Weighting. Ilja Kuzborskij, Claire Vernade, Andras Gyorgy, Csaba Szepesvari Proceedings of The 24th International Conference on Artificial Intelligence and Statistics, PMLR 130:640-648, 2021.
   + PAC-Bayesian Offline Contextual Bandits With Guarantees. Otmane Sakhi, Pierre Alquier, Nicolas Chopin Proceedings of the 40th International Conference on Machine Learning, PMLR 202:29777-29799, 2023.
   + Importance-Weighted Offline Learning Done Right. Germano Gabbianelli, Gergely Neu, Matteo Papini Proceedings of The 35th International Conference on Algorithmic Learning Theory, PMLR 237:614-634, 2024.
   + Logarithmic Smoothing for Pessimistic Off-Policy Evaluation, Selection and Learning. Otmane Sakhi, Imad Aouali, Pierre Alquier, Nicolas Chopin.









[1] Off-Policy Evaluation for Large Action Spaces via Embeddings. Yuta Saito, Thorsten Joachims.

[2] Off-Policy Evaluation for Large Action Spaces via Conjunct Effect Modeling. Yuta Saito, Qingyang Ren, Thorsten Joachims

[3] Rieke, N., Hancox, J., Li, W. et al. The future of digital health with federated learning. npj Digit. Med. 3, 119 (2020). https://doi.org/10.1038/s41746-020-00323-1

**Questions:**

- What are the challenges that make the cross-domain application more difficult/particular compared to the large action space scenario? If it is just deterministic policies/deficient support, then it is not different than the case of large action spaces.
- What is the originality of the bias/variance study of the COPE estimator?
- How can you be sure that your OPL method will converge to the right policy?
- Is there a reason why you omit the whole OPL literature based on pessimism?

---

> ### Author Response · Authors · 2024-11-15
>
> We appreciate the valuable and thoughtful feedback from the reviewer. We respond to the concrete questions and comments in detail below.
>
> > What are the challenges that make the cross-domain application more difficult/particular compared to the large action space scenario?
>
> We would like to address a potential critical misunderstanding by the reviewer. Our motivation is substantially different from that of OPE/L for large action spaces; our main focus is on addressing issues such as extremely small sample sizes, deterministic logging, and new actions in the target domain, rather than to address the variance issue of importance weighting in large action spaces. Moreover, in our setup, we do not assume access to action features or embeddings, making methods developed for OPE/L in large action spaces infeasible. Our work demonstrates that we can still handle these challenging cases by leveraging both target and source domain data in a principled way. If action embeddings are available, our method can even be combined with methods developed for large action spaces, indicating that our formulation and OPE/L for large action spaces are orthogonal to each other.
>
> > If it is just deterministic policies/deficient support, then it is not different than the case of large action spaces.
>
> We respectfully disagree with the reviewer on this point. Our problem setup for cross-domain OPE/L is substantially different from that of OPE/L for large action spaces. First, we do not assume access to action embeddings. We also consider cases with completely new actions in the target domain, making it infeasible to perform representation learning of actions to obtain such embeddings. In addition, as we described above, if multiple domains and action embeddings are both available, it is possible to combine our estimator with approaches developed for large action space problems. This suggests that our approach is orthogonal to methods designed for large action spaces. It seems important for the reviewer to understand that methods developed for large action spaces are simply infeasible in our problem setup.
>
>
> > What is the originality of the bias/variance study of the COPE estimator?
>
> Our analysis is original because we are the first to analyze the statistical properties of an estimator specific to OPE for the cross-domain setup. If the reviewer is aware of any existing paper that improves and analyzes OPE in the cross-domain setup, we would be more than happy to discuss it. If no such paper exists, this would mean that we are the first to propose and analyze a principled approach for leveraging cross-domain data to expand the applicability of OPE/L.
>
> While it is true that our analysis is inspired by existing methods for large action spaces (as we have already mentioned in our paper), no one previously demonstrated that cross-domain data could be leveraged through the high-level decomposition approach. It might seem straightforward after seeing the formulation that we developed, but it is not at all straightforward to come up with it from scratch. This is why it is worth writing a whole new paper to formulate this problem setup and to demonstrate the effectiveness of our approach to leverage cross-domain data from both theoretical and empirical perspectives.
>
> > How can you be sure that your OPL method will converge to the right policy?
>
> We are not certain what the reviewer means by “the right policy” in the question.
>
> If this refers to the “optimal” policy, no methods converge to it in scenarios with deterministic logging and new actions due to severe bias in policy gradient estimation. This is why we empirically compare the effectiveness of off-policy learning methods under those challenging (but prevalent) cases comprehensively. Our experiments demonstrate that our OPL method outperforms existing methods under the challenging scenarios, by leveraging source domain data to reduce bias and variance in estimating the policy gradient for the target domain and for new actions.
>
> > Is there a reason why you omit the whole OPL literature based on pessimism?
>
> **This is simply because we did not aim to study the effectiveness of the pessimism approach in our work**. It is important to understand that our main motivation is to provide an approach that leverages both source and target domain data to perform OPE/L more effectively, even when the target data is extremely limited, involves deterministic logging, or includes new actions. **Pessimism is irrelevant in this context because it does not address the specific challenges we tackle**. However, it is straightforward to incorporate pessimism with our OPL method if one wants to do so in practice. We will clarify this point in the revision.

---

> > ### Comment · Reviewer_KgjZ · 2024-11-20
> >
> > I would like to thank the authors for their answer. Below, I respond to the points raised and to the potential misunderstanding.
> >
> > > our main focus is on addressing issues such as extremely small sample sizes, deterministic logging, and new actions in the target domain, rather than to address the variance issue of importance weighting in large action spaces.
> >
> > The method proposed has the merit to work on the cross-domain application, but all the mentioned issues are addressed by OffCEM [1] (which improves on MIPS [2]) **if we focus on the one-domain application**. OffCEM [1] deals with small sample sizes (as it reduces drastically the variance), deterministic logging policies (as it uses importance sampling on the cluster level, not the action level) and new actions (same, importance sampling on the cluster level). This means that the contribution of the work is to understand that the cross-domain application suffers from the same problems and adapt the OffCEM method to it.
> >
> > > Moreover, in our setup, we do not assume access to action features or embeddings, making methods developed for OPE/L in large action spaces infeasible. Our work demonstrates that we can still handle these challenging cases by leveraging both target and source domain data in a principled way.
> >
> > OffCEM [1] (which improves on MIPS [2]) uses clusters in the same way as the method proposed. These clusters can be defined from the actions themselves (leveraging side information), action/context pairs or even the reward signal itself [3]. I understand the difference between your application and the large action space scenario, but the idea of the decomposition used in OffCEM is general enough to be transfered to other applications. This means that if there is any contribution, it is mostly on the definition/ **the identification of an application that can benefit from this decomposition technique**. Actually, the same decomposition was used multiple times in a multitude of applications [4, 5, 6], because all these applications present the same underlying challenges.
> >
> > > Our analysis is original because we are the first to analyze the statistical properties of an estimator specific to OPE for the cross-domain setup.
> >
> > Indeed, you are the first to analyse an off-policy estimator for the cross-domain setup. But the analysis does not tell us anything about the difficulty of the cross-domain setup, as it is purely based on a decomposition technique used in other applications. For example, what is the bias of **IPS-ALL/DR-ALL** in this setup? and how can we be sure that COPE improves it? Can we prove something similar for OPL?
> >
> > > If this refers to the “optimal” policy, no methods converge to it in scenarios with deterministic logging and new actions due to severe bias in policy gradient estimation.
> >
> > Yes thank you, I was referring to the optimal policy. Policy gradient based on the COPE estimator works well on your semi-synthetic experiments but it is not enough to understand the failure modes of the method. For example, these questions are interesting: When is COPE not expected to work? What is the policy that COPE converges to and when does it match the optimal policy?
> >
> > > This is simply because we did not aim to study the effectiveness of the pessimism approach in our work.
> >
> > > Pessimism is irrelevant in this context because it does not address the specific challenges we tackle
> >
> > I completely get it. I asked about it because your work only talks about PG as a way to learn policies and It reduces the large spectrum of OPL work to PG, while there are other, more principled way to learn policies. Including this line of work in the extended related work can be of relevance to the readers.
> >
> >
> >
> > **- References**
> >
> > [1] Off-Policy Evaluation for Large Action Spaces via Conjunct Effect Modeling. Yuta Saito, Qingyang Ren, Thorsten Joachims
> >
> > [2] Off-Policy Evaluation for Large Action Spaces via Embeddings. Yuta Saito, Thorsten Joachims.
> >
> > [3] Marginal Density Ratio for Off-Policy Evaluation in Contextual Bandits. Muhammad Faaiz Taufiq, Arnaud Doucet, Rob Cornish, Jean-Francois Ton
> >
> > [4] Off-Policy Evaluation of Slate Bandit Policies via Optimizing Abstraction. Haruka Kiyohara, Masahiro Nomura, Yuta Saito
> >
> > [5] Long-term Off-Policy Evaluation and Learning. Yuta Saito, Himan Abdollahpouri, Jesse Anderton, Ben Carterette, Mounia Lalmas
> >
> > [6] Effective Off-Policy Evaluation and Learning in Contextual Combinatorial Bandits. Tatsuhiro Shimizu, Koichi Tanaka, Ren Kishimoto, Haruka Kiyohara, Masahiro Nomura, Yuta Saito

---

> > > ### Author Response · Authors · 2024-11-20
> > >
> > > We appreciate the response from the reviewer, and there are many interesting points to discuss in the responses. We will address them below.
> > >
> > > > The method proposed has the merit to work on the cross-domain application, but all the mentioned issues are addressed by OffCEM [1] (which improves on MIPS [2]) if we focus on the one-domain application.
> > >
> > > > OffCEM [1] deals with small sample sizes (as it reduces drastically the variance), deterministic logging policies (as it uses importance sampling on the cluster level, not the action level) and new actions (same, importance sampling on the cluster level).
> > >
> > >
> > > **We respectfully disagree with the reviewer’s understanding of OffCEM. It appears that the reviewer believes OffCEM can address the issues of deterministic logging and new actions, but this is not correct.**
> > >
> > > First, if the logging policy $\pi_0(a|x)$ is deterministic, the logging cluster distribution $\pi_0(c|x)$ used in the denominator of OffCEM also becomes deterministic. This follows directly from the definition: $\pi_0(c|x) = \sum_a \mathbb{I} \\{ c_a = c \\} \pi_0(a|x)$, where $c_a$ is some action clustering function. When $\pi_0(c|x)$ is deterministic, the common cluster support condition of OffCEM is severely violated, resulting in substantial bias. Our method addresses this issue by leveraging cross-domain data, and this is one of the critical distinctions between our motivation and that of OffCEM.
> > >
> > > Second, if new actions exist in the domain of interest, there is no way for OffCEM to estimate the residual effect for those actions, resulting in unavoidable bias there as well. This is another critical distinction from our contributions, which demonstrate that COPE can address the residual effect and can be unbiased even in the presence of new actions (note that COPE needs not be completely unbiased to outperform the baselines as empirically demonstrated). Furthermore, there is no information or data available about these new actions, making it impossible to perform meaningful clustering for them in the OffCEM setting.
> > >
> > > Moreover, **the terms “deterministic logging” and “new action,” as well as experiments regarding these setups, never appear in the OffCEM paper.** This is indeed reasonable because OffCEM is not designed to solve these issues. As described above, we can demonstrate from a technical perspective that OffCEM and single-domain applications cannot address these settings, and that is why we consider additionally leveraging source domain data to deal with these untackled challenges. We believe understanding this distinction is essential for accurately evaluating our contributions.
> > >
> > >
> > > > OffCEM [1] (which improves on MIPS [2]) uses clusters in the same way as the method proposed. These clusters can be defined from the actions themselves (leveraging side information), action/context pairs or even the reward signal itself [3].
> > >
> > > We would also like to highlight a critical misunderstanding here. First, OffCEM performs clustering in the action space, whereas COPE performs clustering in the domain space. This leads to a critical distinction we discussed earlier, namely that OffCEM cannot (and did not aim to) address deterministic logging and new actions, while COPE addresses these challenges by leveraging cross-domain data. The same applies to the approach in [3]. In the presence of the issues we address, the weight estimation procedure of the method in [3] is severely biased (which needs access to the importance weights as in their Section 3.1.2), just as OffCEM does. This is expected, as all of [1, 2, 3] are designed for dealing with the variance issues in the single-domain setup without determining logging and completely new actions (note that deterministic logging is a harder problem that deficient support).
> > >
> > > Furthermore, the reviewer might have assumed that we rely on the availability of side information, such as action features, but this is not the case in our paper. We consider a scenario where no such action features are available (consistent with typical OPE papers), and there is no way for [1, 2] to perform meaningful clustering under deterministic logging and new actions without such additional information regarding the actions. However, we agree that it is valuable to combine action features and cross-domain data when both are available, but this would not be possible without the development of our formulation and method for the cross-domain setup.

---

> ### Author Response · Authors · 2024-11-20
> **Official Comment by Authors (cont'd)**
>
> > Actually, the same decomposition was used multiple times in a multitude of applications [4, 5, 6], because all these applications present the same underlying challenges.
>
> These are indeed very interesting references! The fact that these papers ([4, 5, 6]) have been accepted by top-tier conferences demonstrates that formulating and solving a new problem using a (seemingly) similar decomposition approach, along with providing relevant empirical demonstrations, is indeed considered a valuable and non-trivial contribution.
>
> It seems the reviewer believes it is trivial to apply the decomposition approach to a different problem setup. If it is indeed true that the reviewer immediately came up with the cross-domain setup upon learning about the decomposition approach, it would be truly impressive and extraordinary. However, this is not the case for many other potential readers, which is one of the reasons why the previous conferences evaluate papers like [4, 5, 6] as worth sharing with the relevant audience. If the reviewer believes that only our contribution is not worth presenting, despite the existence of those multiple similar “types” of contributions that have been accepted as the reviewer mentioned, we would be really eager to learn the reviewer’s reasoning behind this. Such feedback would be valuable in improving our approach to research in general.
>
>
> > But the analysis does not tell us anything about the difficulty of the cross-domain setup, as it is purely based on a decomposition technique used in other applications.
>
> As we have demonstrated above, in the presence of deterministic logging and new actions, the existing decomposition approach of OffCEM cannot be unbiased or address the residual effect in the off-policy setup. Our bias analysis indicates that the residual effect can be addressed depending on the pairwise accuracy of $\hat{q}$ and that it can be unbiased under Condition 3.2. This is in contrast to OffCEM, which cannot achieve unbiasedness under any additional conditions in the presence of deterministic logging and new actions. Therefore, our analysis provides new insights that did not appear in the previous papers, and thus we do not think that our analysis “purely” relies on previous results.
>
> It is also important to clarify that we do not necessarily aim to highlight the “difficulty” of the cross-domain setup. Instead, we regard source domain data as "additional" information that can potentially be leveraged to address the already challenging issues. Nevertheless, we can still discuss the difficulty of the cross-domain setup, particularly as it relates to the varying distributions across domains, as we will elaborate on in the following section.
>
>
> > For example, what is the bias of IPS-ALL/DR-ALL in this setup? and how can we be sure that COPE improves it?
>
> We can derive the bias of IPS-ALL as follows (we will include the detailed derivation and the bias of DR-ALL, which is a straightforward extension, in the revision).
>
> $Bias (\hat{V}\_{\mathrm{IPS}} (\pi ; \mathcal{D})) = | \mathbb{E}_{p_T(x)\pi(a|x)}[q^T(x,a)] -  \sum\_k \frac{n^k}{N} \mathbb{E}\_{p_k(x)\pi(a|x)}[q^k(x,a)]  |$
>
> We observe that it is no longer guaranteed to remain unbiased in our setup. This bias becomes significant when the reward distributions across domains differ substantially because the expectation of IPS-ALL is characterized by the weighted sum of the policy values of all the domains involved. In contrast, the bias becomes zero when the reward distributions across all domains are identical (which is no longer a cross-domain setup).
>
> Based on this derivation and Theorem 3.1, we can compare the bias of COPE and that of IPS/DR-ALL as follows:
> - COPE performs better when the distributions across domains are substantially different and also when the pairwise estimation by $\hat{q}$ is accurate.
>
> - IPS/DR-ALL performs better when the distributions across domains are similar, i.e., when the cross-domain setup is close to the typical single-domain setup.
>
> When the setup is close to the typical single-domain scenario, effective estimators such as DR and its extensions are already well-known. Therefore, we focused more on novel and challenging scenarios where the distributions across domains are not necessarily similar, as baseline methods tend to fail in these cases as shown above and in our experiments.
>
> In practice, we do not precisely know which of COPE and baselines are better. This is not the case only for our setup, but also for the other OPE problems. A typical example is that we never know which of IPS and DM is better, because the comparison depends on many (potentially unknown) parameters such as the reward noise. This is why there exists an orthogonal line of work around estimator selection in OPE [7, 8, 9, 10], and we can use those methods to identify the appropriate estimator tailored to the specific problem when we use OPE in general and in the cross-domain setup.

---

> > ### Author Response · Authors · 2024-11-20
> > **Official Comment by Authors (cont'd)**
> >
> > [7] Yi Su, Pavithra Srinath, Akshay Krishnamurthy. Adaptive Estimator Selection for Off-Policy Evaluation. ICML2020.
> >
> > [8] Allen Nie, Yash Chandak, Christina J. Yuan, Anirudhan Badrinath, Yannis Flet-Berliac, Emma Brunskil. OPERA: Automatic Offline Policy Evaluation with Re-weighted Aggregates of Multiple Estimators. NeurIPS2024.
> >
> > [9] Takuma Udagawa, Haruka Kiyohara, Yusuke Narita, Yuta Saito, Kei Tateno. Policy-Adaptive Estimator Selection for Off-Policy Evaluation. AAAI2023.
> >
> > [10] Nicolò Felicioni, Michael Benigni, Maurizio Ferrari Dacrema. Automated Off-Policy Estimator Selection via Supervised Learning. https://arxiv.org/abs/2406.18022
> >
> > > I was referring to the optimal policy. Policy gradient based on the COPE estimator works well on your semi-synthetic experiments but it is not enough to understand the failure modes of the method. For example, these questions are interesting: When is COPE not expected to work? What is the policy that COPE converges to and when does it match the optimal policy?
> >
> > Thank you for the clarification. Regarding the failure modes, we can use the discussions above comparing COPE and IPS/DR-ALL to address this question. Specifically, COPE may perform worse relative to IPS/DR-ALL when the distributions across domains become similar (i.e., a setup closer to the single-domain setup where we already know many effective estimators). Another failure mode arises when the regression model $\hat{q}$ is drastically inaccurate. However, such inaccuracy would also significantly degrade the performance of DR, leaving IPS as the only method likely to perform reasonably well in this setup.
> >
> > Regarding the second question, COPE can converge to the optimal policy under the satisfaction of the conditional pairwise correctness (CPC) stated in Condition 3.2. However, we do not expect CPC to hold in practice (we included it merely to provide a theoretical understanding of when COPE can be unbiased, and the condition is not necessary for COPE to outperform the baselines as already demonstrated). Nevertheless, we can still analyze the convergence of COPE under satisfied CPC. This analysis builds on the well-known convergence analysis for OPL presented such as in [11, 12], and we can derive that COPE converges to the optimal policy at a rate of $\mathcal{O}(\sqrt{V/n})$, where $n$ is the training data size and $V$ is the (asymptotic) variance of COPE. If the reviewer believes this additional analysis would add value, we will include it in the revised version.
> >
> > [11] Masahiro Kato, Masatoshi Uehara, Shota Yasui. Off-Policy Evaluation and Learning for External Validity under a Covariate Shift. NeurIPS2020.
> >
> > [12] Zhengyuan Zhou, Susan Athey, Stefan Wager. Offline Multi-Action Policy Learning: Generalization and Optimization. https://arxiv.org/abs/1810.04778
> >
> > > I completely get it. I asked about it because your work only talks about PG as a way to learn policies and It reduces the large spectrum of OPL work to PG, while there are other, more principled way to learn policies. Including this line of work in the extended related work can be of relevance to the readers.
> >
> > Yes, we will include a discussion of the line of work on pessimism in the related work section. We sincerely appreciate the reviewer for providing these references once again.

---

> ### Comment · Reviewer_KgjZ · 2024-11-20
>
> I thank the authors for their thorough answer, It really helped me understand the work more. I will try to keep my response brief.
>
> - Thank you for clarifying the differences between OffCEM and COPE. Indeed, even if the decomposition is similar, and that OffCEM can deal with some challenges in the **one domain application** (variance issues and deficient support), COPE uses this decomposition differently (with clustering on the domain) and tackles more severe challenges (deterministic logging policy on the target domain).
>
> - **Limitations** of the approach should be discussed openly, for instance, these severe challenges (deterministic logging policy on the target domain) can only be tackled in the presence of nice data from other domains (stochastic policies in other domains for instance). It should also be highlighted that clustering the average reward will not always work if the average reward on all domains is the same, but the distribution of $r(a,x)$ is different in all domains. Another limitation that was not raised before, is the estimation of $p^{\phi(T)}(x, a)$ that involves ratios of covariate densities $p^k(x)$. The synthetic experiments conducted assume access to $p^k(x)$, which is far from practice. Indeed, these densities are hard to model in real world scenarios and estimating them will bias the approach, meaning that adopting this method in practice will require substantial effort and will always suffer from additional modelling bias.
>
>
> - Pointing to [4, 5, 6] was to show that a lot of works are adopting these techniques and novelty comes from the application more than the technique itself, actually, [4, 5, 6] were all published in applied conferences. If the paper is positioned as applied work, it is fine, but I deem that the paper needs more challenging experiments than the synthetic ones to properly validate the approach.
>
> - Thank you for the developed bias discussion, it may be added to the appendix. I think it can be of benefit to readers. A rigorous theoretical comparison between estimators in the cross domain setting will require another paper.
>
> - **Questions :**
>
>    - Did you try harder experiments where you do not have access to $p^k(x)$, and experiments with really difficult $p^k(x)$ to model?

---

> > ### Author Response · Authors · 2024-11-21
> >
> > We appreciate the prompt yet thoughtful response from the reviewer. We would like to discuss some of the comments in detail below.
> >
> > > these severe challenges (deterministic logging policy on the target domain) can only be tackled in the presence of nice data from other domains (stochastic policies in other domains for instance)
> >
> > We believe it is always the case that “nice data” are required to perform reliable estimation, not only in our problem but also in any statistical estimation problem. For this reason, we did not mention it in our paper to avoid redundancy.
> >
> > What is important to remember here is that, without our formulation and method, there is no established way to leverage cross-domain data in OPE/L, even if the data are indeed informative. However, if the reviewer believes that mentioning this as a limitation of our work is indeed useful, we can revise the paper accordingly during the discussion period.
> >
> > > It should also be highlighted that clustering the average reward will not always work if the average reward on all domains is the same, but the distribution of r(a,x) is different in all domains.
> >
> > This is an interesting point to discuss. First, regarding the clustering procedure that we implemented in our experiments, **we have already stated the following in Section 6**
> >
> > - *even though the heuristic domain clustering of using the empirical averaged rewards as domain embeddings worked satisfactorily in our experiments, it would be valuable to develop a more principled method to perform clustering of domains*
> >
> > If the reviewer thinks that this is insufficient, we can also specify this as a potential limitation, which is immediately revisable. We believe that developing a principled and data-driven clustering procedure would be a highly intriguing topic on its own and would require another full paper (like in the literature about large action spaces where there exist multiple papers focusing on action clustering or representation learning).
> >
> >
> > > Another limitation that was not raised before, is the estimation of pϕ(T)(x,a) that involves ratios of covariate densities pk(x) . The synthetic experiments conducted assume access to pk(x) , which is far from practice. Indeed, these densities are hard to model in real world scenarios and estimating them will bias the approach, meaning that adopting this method in practice will require substantial effort and will always suffer from additional modelling bias.
> >
> > Thank you for pointing this out; it is indeed worth clarifying. **We did NOT use the true densities $p^{\phi(T)}(x,a)$ or $p^k(x)$ at all in our experiments when implementing COPE**. Instead, we performed density ratio estimation to estimate the weights ($\pi\left(a_{i}^{k} \mid x_{i}^{k}\right) / p^{\phi(T)}\left(x_{i}^{k}, a_{i}^{k}\right)$) from only observable logged data, using the implementation provided here: https://github.com/hoxo-m/densratio_py. This can be seen in the code we provided, but we will further emphasize this in the text.
> >
> > Since all experimental results are performed under realistic scenarios with estimated densities, we believe one of the main concerns raised by the reviewer is now addressed.
> >
> >
> > > I deem that the paper needs more challenging experiments than the synthetic ones to properly validate the approach.
> >
> > It seems not entirely clear what the reviewer means by “more challenging experiments.”
> >
> > We would like to point out that the papers mentioned by the reviewer, such as [4, 5], rely on synthetic data relevant to their setup as well as the KuaiRec dataset, exactly the same dataset we used. The KuaiRec dataset is the only publicly available dataset that provides the full reward matrix, a crucial requirement for conducting experiments related to OPE. We can indeed provide dozens of papers accepted at ICLR or other relevant conferences that use similarly or even more synthetic experiments (including [1, 2, 3, 7, 8, 9, 10, 11, 12]). We agree that it would be ideal to construct more real-world public datasets that enable rigorous experiments on OPE and OPL as a community.
> >
> > We believe we have followed the best practices given the currently available resources. And given the above context, we would like to seek clarification on what the reviewer specifically means by “more challenging experiments”.

---

> > > ### Comment · Reviewer_KgjZ · 2024-11-21
> > >
> > > I would like to thank the authors for their answer.
> > >
> > > - It is really important to talk about how the density ratios were estimated, at it is fundamental to the estimator proposed and cannot be avoided. Can you please elaborate more on this method?
> > >
> > > - By challenging experiments, I mean experiments that do not favour the COPE estimator. For instance, the $p^k(u)$ used in your experiments are defined using the true average reward of the user, so depending on the values of $\alpha^k$ used, you'll obtain domains with more 'active' users, which plays favourably to your reward clustering approach.
> > >    - More challenging experiments should be experiments where we do not have control over $p^k(u)$. For example, one can imagine an experiment where we have labeled documents. The action space would be the label space and the domains can be defined by some metadata of the documents. This metadata should only be used to define the domains and is not accessible to the practitioner. We can then use the supervised to bandit conversion to simulate the cross-domain off-policy estimation/learning problem.

---

> > > > ### Author Response · Authors · 2024-11-21
> > > >
> > > > We would like to thank the reviewer for the additional comments and suggestions.
> > > >
> > > > > It is really important to talk about how the density ratios were estimated, at it is fundamental to the estimator proposed and cannot be avoided. Can you please elaborate more on this method?
> > > >
> > > > Yes, we are absolutely happy to clarify. Our implementation in the experiments relies on one of the most standard methods, unconstrained Least-Squares Importance Fitting (uLSIF), to perform density ratio estimation proposed in [13].
> > > >
> > > > The method considers minimizing the following squared error between $s$ and $r$ where $s$ is an estimator for the ratio and $r(x,a) = \pi(a| x)/p^{\phi(T)}(x, a)$ is the ratio we aim to estimate:
> > > >
> > > > $$\mathbb{E} [(s(x,a)-r(x,a))^{2}]=\mathbb{E}\_{\pi(a|x)} [(r(x,a))^{2}]-2 \mathbb{E}\_{p^{\phi(T)}(a|x)} [s(x,a)]+\mathbb{E}\_{\pi(a|x)} [(s(x,a))^{2}]$$
> > > >
> > > > The first term of the equation does not affect the result and thus we can ignore it as follows:
> > > >
> > > > $$s^{*}=\underset{s \in \mathcal{S}}{\arg \min }[\frac{1}{2} \mathbb{E}\_{p^{\phi(T)}(a|x)} [(s(x,a))^{2}]-\mathbb{E}\_{\pi(a|x)} [s(x,a)]]$$
> > > >
> > > > To solve the empirical version of this optimization, [13] use kernel based hypotheses to estimate the density ratio nonparametrically with a regularization term as usual. https://github.com/hoxo-m/densratio_py is one of the well-known public implementations of the method. We will summarize this in the appendix.
> > > >
> > > > [13] Kanamori, T., Suzuki, T., and Sugiyama, M. Statistical analysis of kernel-based least-squares density-ratio estimation. Mach. Learn., 86(3):335–367, 2012.
> > > >
> > > > > By challenging experiments, I mean experiments that do not favour the COPE estimator. For instance,
> > > >
> > > > Thank you for the valuable suggestion. We agree that investigating the robustness of COPE to different $p^k(x)$ is an interesting ablation. This analysis can be conducted not only on classification data but also on the KuaiRec dataset, as in any case, we need to define the true underlying $p^k(x)$ when generating the data. We will attempt to perform such an ablation study during the discussion phase.
> > > >
> > > > In the meantime, we would like to clarify the remaining concerns of the reviewer. It seems to us that the reviewer’s only concern at the moment is the need for the additional ablation study. We find it a little surprising that this was never mentioned in the initial review as a reason to reject. Indeed, the reviewer evaluated our experimental section as one of the strengths. While we understand that the reviewer’s evaluation can change during the discussion phase (which is the point of it), we would like to confirm whether this additional experiment is the only remaining concern preventing a change in the reviewer’s score.
> > > >
> > > > If the reviewer has any other remaining concerns, we would be eager to learn them to further improve the paper.

---

> > > > > ### Comment · Reviewer_KgjZ · 2024-11-21
> > > > >
> > > > > Thank you for elaborating the density ratio estimation technique used. It is really important to talk about it in the appendix.
> > > > >
> > > > > It is true that my criticism at the beginning was directed towards the originality of the theoretical contributions of the paper, because if the goal is a theoretical contribution, synthetic experiments to validate the approach are enough. But I understood through the rebuttal that the paper is intentionally positioned as an applied paper. This means that the paper needs stronger experiments to validate the method proposed, which is not the case in the current version as the method is only validated on synthetic experiments in settings that favour the approach proposed. Stronger experiments will definitely make me raise my score.

---

> > > > > > ### Author Response · Authors · 2024-11-22
> > > > > >
> > > > > > Thank you so much for clarifying the reviewer’s current evaluation (which we think is reasonable and convincing) and for the effort put into thoroughly understanding our work.
> > > > > >
> > > > > > With our current implementation, it is not very difficult to extend the experimental setup as suggested, and we will follow up if we obtain additional results to share in time.

---

> > > > > > > ### Author Response · Authors · 2024-11-26
> > > > > > >
> > > > > > > We appreciate the timely and thoughtful discussion provided by the reviewer once again.
> > > > > > >
> > > > > > > We would now like to share additional experimental results obtained from the ablation study we performed based on the reviewer’s suggestion. Specifically, we exclude $\bar{q}(u)$ from the definition of the user distribution $p^k(u)$, following the reviewer’s observation that this might favor our methods. Instead, we define a linear function of the user feature $f^k(u) = \beta^k x_u$, where the coefficient vector $\beta^k$ is sampled from a normal distribution separately for each domain $k$. We then define the user distribution $p^k(u)$ for each domain as $ p^{k}(u):=\frac{\exp (f^k(u))}{\sum_{u^{\prime} \in \mathcal{U}} \exp (f^k(u^{\prime}))} $. This procedure follows the multi-domain recommendation literature, which uses user features to define the domains in their experiments [Yang et al. 2024]. More importantly, this approach avoids any dependence of the distribution on $\bar{q}(u)$. We conducted both OPE and OPL experiments based on this new user distribution, and the results are summarized in the following tables.
> > > > > > >
> > > > > > > ---
> > > > > > > **MSE in OPE for varying ratios of new actions in the target domain** (values are smaller the better)
> > > > > > > | ratios of new actions | 0.0    | 0.2    | 0.4    | 0.6    | 0.8    |
> > > > > > > |-----------|--------|--------|--------|--------|--------|
> > > > > > > | **COPE(Ours)**       | **1.1641** | **1.1647** | **0.8066** | **1.0663** | **1.0801** |
> > > > > > > | DR(Target Domain)        | 4.3283 | 4.2225 | 2.3985 | 2.9498 | 3.7964 |
> > > > > > > | DR(ALL Domain)    | 2.7582 | 2.2401 | 3.5354 | 3.3172 | 1.4530 |
> > > > > > > | IPS(Target Domain)       | 5.6442 | 4.9797 | 3.6800 | 4.8295 | 6.5302 |
> > > > > > > | IPS(ALL Domain)   | 2.1763 | 2.2538 | 2.2006 | 2.1881 | 2.9360 |
> > > > > > >
> > > > > > > **MSE in OPE for varying numbers of users with deterministic logging in the target domain** (values are smaller the better)
> > > > > > >
> > > > > > > | numbers of users with deterministic logging | 200    | 400    | 600    | 800    | 1000   |
> > > > > > > |-----------|--------|--------|--------|--------|--------|
> > > > > > > | **COPE(Ours)**      | **1.3238** | **1.3120** | **1.2926** | **1.2125** | **1.2172** |
> > > > > > > | DR(Target Domain)        | 2.3200 | 2.8398 | 3.2755 | 3.7992 | 4.4499 |
> > > > > > > | DR(ALL Domain)    | 2.5240 | 2.7915 | 3.0358 | 2.9843 | 3.3314 |
> > > > > > > | IPS(Target Domain)       | 3.2026 | 3.9641 | 4.6112 | 5.6947 | 7.0049 |
> > > > > > > | IPS(ALL Domain)   | 2.1561 | 2.1630 | 2.1906 | 2.2188 | 2.2076 |
> > > > > > >
> > > > > > >
> > > > > > > **Test (Relative) Policy Value in OPL for varying ratios of new actions in the target domain** (values are larger the better)
> > > > > > > | ratios of new actions    | 0.0    | 0.2    | 0.4    | 0.6    | 0.8    |
> > > > > > > |--------------|--------|--------|--------|--------|--------|
> > > > > > > | **COPE-PG(Ours)**      | **1.4463** | **1.4060** | **1.4389** | **1.3884** | 1.4090 |
> > > > > > > | DR-PG(Target Domain)        | 1.3518 | 1.3220 | 1.3364 | 1.2663 | 1.3339 |
> > > > > > > | DR-PG(ALL Domain)    | 1.4308 | 1.3947 | 1.4064 | 1.3420 | **1.4403** |
> > > > > > > | IPS-PG(Target Domain)        | 1.3119 | 1.2755 | 1.2899 | 1.2389 | 1.3350 |
> > > > > > > | IPS-PG(ALL Domain)   | 1.3588 | 1.3198 | 1.3267 | 1.3134 | 1.3246 |
> > > > > > >
> > > > > > >
> > > > > > > **Test (Relative) Policy Value in OPL for varying numbers of users with deterministic logging in the target domain** (values are larger the better)
> > > > > > > | numbers of users with deterministic logging    | 200    | 400    | 600    | 800    | 1000   |
> > > > > > > |--------------|--------|--------|--------|--------|--------|
> > > > > > > | **COPE-PG(Ours)**      | **1.4408** | **1.4531** | **1.4769** | **1.4669** | 1.4139 |
> > > > > > > | DR-PG(ALL Domain)   | **1.4477** | 1.4180 | 1.3954 | 1.4059 | **1.5115** |
> > > > > > > | DR-PG(Target Domain)     | 1.3101 | 1.3014 | 1.3091 | 1.3596 | 1.3013 |
> > > > > > > | IPS-PG(ALL Domain)  | 1.3381 | 1.3003 | 1.3149 | 1.3625 | 1.4651 |
> > > > > > > | IPS-PG(Target Domain)    | 1.2795 | 1.2770 | 1.2905 | 1.2765 | 1.2467 |
> > > > > > >
> > > > > > > ---
> > > > > > >
> > > > > > > **The results indicate that our method mostly outperforms the baseline methods that use only the target domain data or all the data across target and source domains in both OPE and OPL experiments** (bold fonts indicate the best method, if the best and second best are not significantly different, we use bold fonts for both).

---

> ### Author Response · Authors · 2024-11-26
>
> Furthermore, we can extend the definition of the user distribution as:
>
> $$ p^{k}(u):=\frac{\exp (\lambda \bar{q}(u) + (1-\lambda) f^k(u))}{\sum_{u^{\prime} \in \mathcal{U}} \exp (\lambda \bar{q}(u’) + (1-\lambda) f^k(u')))} $$
>
> where $\lambda$ controls the degree to which the user distribution depends on $\bar{q}(u)$. When $\lambda = 1$, the setup reduces to our original experiment. The following reports the OPL experiment with varying $\lambda$ within $\\{ 0, 0.2, 0.4, 0.6, 0.8, 1.0 \\}$.
>
>
> **Test (Relative) Policy Value in OPL for varying $\lambda$ values** (values are larger the better)
> | lambda values   | 0.0   | 0.2   | 0.4   | 0.6   | 0.8   | 1.0   |
> |------------|-----------|-----------|-----------|-----------|-----------|-----------|
> | **COPE(Ours)**  | **1.4713**    | **1.4646**    | **1.4739**    | **1.4502**    | **1.4780** | **1.4404** |
> | DR-PG(Target Domain) | 1.3812    | 1.3558    | 1.3475    | 1.3592    | 1.3042    | 1.3657    |
> | DR-PG (ALL Domain) | 1.4639    | 1.4334    | 1.3823    | 1.4374    | 1.3956    | 1.2799    |
> | IPS-PG(Target Domain)| 1.3553    | 1.3010    | 1.3133    | 1.2869    | 1.2770    | 1.2755    |
> | IPS-PG (ALL Domain) | 1.3629    | 1.3580    | 1.3572    | 1.3480    | 1.4390    | 1.3889    |
>
> We should mention here that the difference between the proposed method and the baselines decreases when $\lambda$ is small, which validates the reviewer’s comment. **These additional observations, however, demonstrate the robustness of COPE to varying user distributions, even with a simple clustering procedure.** Nevertheless, we believe a more principled and effective approach to domain clustering for our method likely exists. As mentioned in Section 5, we consider the development of a clustering algorithm tailored to Cross-Domain OPE and OPL to be a valuable direction for future research.
>
> As a side note, we have updated our draft to include clarifications on points discussed with the reviewers, such as data-driven tuning of cluster numbers, bias comparisons against IPS/DR-ALL, density ratio estimation, and the convergence properties of COPE-PG under the CPC condition. We have also clarified that “deterministic logging” and “new actions” can be interpreted as harder versions of the common support violation to avoid confusion.   These updates are highlighted in red in the text. **We would greatly appreciate it if the reviewer could confirm whether the additional results address their main concern or if there are any remaining issues that require further discussion.**
>
> ---
>
> [Yang et al. 2024] Zhiming Yang, Haining Gao, Dehong Gao, Luwei Yang, Libin Yang, Xiaoyan Cai, Wei Ning, Guannan Zhang. MLoRA: Multi-Domain Low-Rank Adaptive Network for CTR Prediction. RecSys2024.

---

> > ### Comment · Reviewer_KgjZ · 2024-11-26
> >
> > Thank you for the timely experiments, they are indeed interesting.
> >
> > If we can summarise the obtained results:
> >
> > - In OPE, COPE always reduces the MSE, even with a naive clustering algorithm.
> > - in OPL, overall, COPE outperforms the baselines with DR-PG (All domains) coming really close, and sometimes outperforming it.
> >
> > These results unveil the importance of the clustering approach and are an invitation to invest in this direction for future research. Can you please add these results and discussions to the paper? **Having them in the appendix and pointing to them from the main body is enough**.
> >
> > I'm satisfied with these new results and discussions in the paper, and I am increasing my score to reflect this. I sincerely hope that the rebuttal phase was beneficial for the work and its authors.

---

> > > ### Author Response · Authors · 2024-11-27
> > >
> > > We appreciate the prompt response from the reviewer. Indeed, this has been one of the best experiences we have had interacting with a reviewer at any conference.
> > >
> > > We agree with the reviewer’s understanding of our additional results, and clustering quality does have some impact on the COPE’s effectiveness (particularly when we use it for OPL). We will summarize the results in the appendix and mention it in the main text. We will then update the draft by the deadline. We also agree with the reviewer in that the development of a clustering algorithm tailored to Cross-Domain OPE and OPL would be a very interesting future topic (as we have mentioned in Section 5).
> > >
> > > We would like to thank the reviewer for their effort in rigorously understanding and improving our contributions once again.

---

### Official Review · Reviewer_NVrf · 2024-11-04

**Soundness:** 2
**Presentation:** 3
**Contribution:** 2
**Rating:** 6
**Confidence:** 2

**Summary:**

Cross-Domain Off-Policy Evaluation and Learning (OPE/L) addresses challenges in contextual bandits, such as few-shot data, deterministic logging, and new actions, by leveraging logged data from multiple domains to evaluate and learn new policies more effectively. This paper introduces a new estimator and policy gradient method that improves OPE/L performance, both theoretical analysis and experimental results are presented.

**Strengths:**

1. This paper studies an interesting and practical problem, with both theoretical analysis and experimental results.
2. This paper is mostly clearly written, and most parts are easy to understand.
3. A suite of experimental results are provided after the theoretical analysis.

**Weaknesses:**

1. I feel the theoretical results are a little bit weak:

1a I am curious about the condition 3.2, is this a common and reasonable assumption? It is better to provide more insightful comments on when this assumption will hold in practice.

1b It is hard for me to understand the paragraph in line 314-320. Why is the estimator unbiased when its cluster size is equal to 1? Intuitively, more clusters in the target cluster $\phi(T)$ mean more logged data come from the same cluster as the new data in the target domain, which will help reduce the bias? Please expand the discussion on this paragraph.

1c I also feel curious about using the off-the-shelf clustering algorithm based on empirical average. How does it work in your theoretical analysis? If the rewards are not very symmetrically distributed, i.e. heavy-tailed distributed, then the empirical mean will lead to terrible estimation, so how to deal with this issue in practice?

1d is there a final regret bound in this paper in terms of the magnitude of $T$ like other bandit works. e.g. $O(\sqrt{T})$

2. Since this paper studies a very niche area of contextual bandits, it is better to give more real-world applications to help readers understand its significance.

3. It is better to report the running time of the proposed algorithm along with baselines to validate the efficiency of the proposed method.

Remark: I checked the codes in the supplementary material and most notes are written in non-English (I guess it is Japanese). Please use English for everything in your source codes, since other languages may lead to violation of double-blind reviewing policy.

**Questions:**

Please refer to the above Weaknesses section.

---

> ### Author Response · Authors · 2024-11-15
>
> We appreciate the valuable and thoughtful feedback from the reviewer. We respond to the concrete questions and comments in detail below.
>
> > 1a I am curious about the condition 3.2, is this a common and reasonable assumption? It is better to provide more insightful comments on when this assumption will hold in practice.
>
> We would like to clarify that we do not expect Condition 3.2 to hold in practice. It is simply a condition that helps to understand when COPE can be unbiased. This is why we first present Theorem 3.1, which characterizes the bias of COPE without assuming Condition 3.2. It is also important to note that in challenging cases, such as deterministic logging and the presence of new actions, every existing estimator, including IPS and DR, produces much bias, but our estimator leverages source domain data and is much more robust to the bias arising due to those challenges. We conducted comprehensive experiments to compare the MSE, bias, and variance of the estimators across a range of challenging cases, and we demonstrated that our estimator performed best in most of such scenarios. **This empirically demonstrates the effectiveness of leveraging cross-domain data through our approach even with an estimated reward function ($\hat{q}$) that likely violates Condition 3.2**.
>
> > 1b It is hard for me to understand the paragraph in line 314-320. Why is the estimator unbiased when its cluster size is equal to 1?
>
> We are happy to clarify this point. When the cluster size is 1, the CPC condition becomes less stringent, resulting in COPE producing a smaller bias. The CPC condition requires that a regression model $\hat{q}(x,a)$ preserve the relative reward differences between different domains within the target cluster (as described in L291–299). When there is only one domain in the target cluster, CPC holds by definition, as the relative reward difference is always zero within a single cluster. In contrast, when there are many domains in the target cluster, the regression model must accurately estimate the relative reward differences for numerous domain pairs, making it more challenging to satisfy this condition, thereby producing some bias.
>
> > 1c I also feel curious about using the off-the-shelf clustering algorithm based on empirical average. How does it work in your theoretical analysis? If the rewards are not very symmetrically distributed, i.e. heavy-tailed distributed, then the empirical mean will lead to terrible estimation, so how to deal with this issue in practice?
>
> It is important to note that our theoretical analysis, including Theorem 3.1, is agnostic to the clustering method; the analysis applies to any given clustering function. Theorem 3.1 suggests that when the relative reward differences between different domains within the target cluster are accurately estimated by $\hat{q}$, the bias of COPE becomes small. Therefore, an effective clustering method is the one that leads to a better estimation of the relative reward differences within the target cluster for a given $\hat{q}$.
>
> In our current experiments, we applied KMeans to the empirical average of the rewards in each domain to perform clustering, and this heuristic worked effectively and was sufficient to outperform the existing methods across a range of experimental setups.
> In practice, the number of clusters and the clustering function used can be considered hyperparameters of COPE. We can tune these hyperparameters rigorously by using estimator selection methods for OPE proposed in the following papers.
>
> - Yi Su, Pavithra Srinath, and Akshay Krishnamurthy. Adaptive estimator selection for off-policy evaluation. ICML2020.
> - Takuma Udagawa, Haruka Kiyohara, Yusuke Narita, Yuta Saito, and Kei Tateno. Policy-adaptive estimator selection for off-policy evaluation. AAAI2023.
>
> We will clarify this point further in the revision.
>
>
> > 1d is there a final regret bound in this paper in terms of the magnitude of $T$ like other bandit works.
>
> We would like to address a potential critical misunderstanding by the reviewer. Our formulation is for an offline contextual bandit, not an online bandit, so regret bound analysis is indeed irrelevant to our work.

---

> > ### Comment · Reviewer_NVrf · 2024-11-19
> > **Thank you for your response**
> >
> > Thank you very much for your responses to my questions. I have read your work again with the explanation in your rebuttal, and now I have a better understanding of your contributions in this work. I have raised my rating with a relatively low confidence, since I am not familiar with the existing literature on this problem setting.

---

> > > ### Author Response · Authors · 2024-11-20
> > >
> > > We appreciate the response from the reviewer and their effort to thoroughly understand our contributions. We believe that we have now addressed all the questions raised by the reviewer in the initial review. Therefore, we wonder what remaining concerns prevent the reviewer from raising the score to "accept." We would be more than happy to discuss any additional concerns during the discussion period.

---

### Meta-Review · Area_Chair_qVq7 · 2024-12-17

**Metareview:**

This paper studies off-policy evaluation and learning in a cross-domain scenario, where logged data from multiple domains can be used to improve the performance in the target domain. The authors propose practical algorithms, analyze them, and evaluate them empirically. The scores of this paper are 8 and 3x 6, which is a major improvement over the initial 6, 2x 5, and 3. This paper was thoroughly discussed in an exemplary discussion. The reviewers had many questions about prior work, technical details, and limited experiments; all of which were answered in detail in the discussion. Therefore, I am confident and glad to recommend this paper for acceptance.

I have one additional comment. The problem of multi-task off-policy learning was studied in [Multi-Task Off-Policy Learning from Bandit Feedback](https://proceedings.mlr.press/v202/hong23a.html). This setting is similar because each task can be viewed as a domain. Please look at this work and the papers that cite it for additional references. Congratulations!

**Additional Comments On Reviewer Discussion:**

See the meta-review for details.

---

### Decision · Program_Chairs · 2025-01-22

Accept (Poster)